# MamKO: Mamba-based Koopman operator for modeling and predictive control

**Zhaoyang Li[1], Minghao Han[2], Xunyuan Yin[1,2,*]**
1 School of Chemistry, Chemical Engineering and Biotechnology, Nanyang Technological University
2 Nanyang Environment and Water Research Institute, Nanyang Technological University
`LIZH0082@e.ntu.edu.sg; minghao.han@ntu.edu.sg; xunyuan.yin@ntu.edu.sg`

## Abstract

The Koopman theory, which enables the transformation of nonlinear systems into linear representations, is a powerful and efficient tool to model and control nonlinear systems. However, the ability of the Koopman operator to model complex systems, particularly time-varying systems, is limited by the fixed linear state-space representation. To address the limitation, the large language model, Mamba, is considered a promising strategy for enhancing modeling capabilities while preserving the linear state-space structure. In this paper, we propose a new framework, the Mamba-based Koopman operator (MamKO), which provides enhanced model prediction capability and adaptability, as compared to Koopman models with constant Koopman operators. Inspired by the Mamba structure, MamKO generates Koopman operators from online data; this enables the model to effectively capture the dynamic behaviors of the nonlinear system over time. A model predictive control system is then developed based on the proposed MamKO model. The modeling and control performance of the proposed method is evaluated through experiments on benchmark time-invariant and time-varying systems. The experimental results demonstrate the superiority of the proposed approach. Additionally, we perform ablation experiments to test the effectiveness of individual components of MamKO. This approach unlocks new possibilities for integrating large language models with control frameworks, and it achieves a good balance between advanced modeling capabilities and real-time control implementation efficiency.

## 1 Introduction

Deep learning methods have made promising achievements in modeling large and complex systems with collected datasets (Wang et al., 2024). The time-series prediction tasks can be well achieved with existing learning technologies and applied to various fields, including weather forecasting (Verma et al., 2024) and traffic prediction (Li et al., 2023). These data-driven methods, especially for deep learning methods, have proven to be highly capable of modeling dynamic systems and provide strong support for advanced control applications.

The Koopman operator theory (Koopman, 1931) has emerged as a promising data-driven method in recent years. The Koopman operator can represent the dynamics of nonlinear systems by linear state-space model (SSM) (Kalman, 1960) by lifting the states from the original space to a higher-dimensional space through a series of observable functions. Based on the theory, extended dynamic mode decomposition (EDMD) (Williams et al., 2015) with manually selected observable functions is proposed. With the advent of deep learning, neural networks (NNs) have been employed to serve as the observable functions in the Koopman operator framework. Advanced deep learning methods, such as autoencoders (Takeishi et al., 2017), probabilistic NNs (Han et al., 2021), and graph NNs (Li et al., 2020), have been integrated with the Koopman operator to enhance the prediction accuracy of the model. The linear controller based on the Koopman model, including linear quadratic regulator (Han et al., 2023), can be efficiently designed from the concise linear model. Although the Koopman model mentioned above shows relatively accurate prediction results, due to the parameter capacity

---

[*]Corresponding author: Xunyuan Yin.

of the Koopman framework, challenges still exist in modeling real-world systems, especially for time-varying systems.

Large language models (LLMs) (Zhao et al., 2023) are deep learning models that leverage a large-scale number of parameters to comprehend and generate human language. The internal architecture of the model enables it to link and reason through contextual information. The most typical structure inside the large language model is Transformer (Wolf et al., 2020). Although the nonlinear function and attention mechanism (Vaswani et al., 2017) primarily improve the approximation capacity, the intricate structures pose challenges for model-based controller design. The Mamba structure (Gu & Dao, 2023) based on SSM is developed for LLMs. Within the structure of Mamba, the similar linear SSM from Koopman operator theory is interpreted as a particular form that integrates features from both recurrent neural networks (RNNs) and convolutional neural networks (CNNs) (Dao & Gu, 2024). Selection mechanisms from Mamba are implemented to improve the modeling ability of linear SSM by generating matrix sequences, which can also be a potential solution for improving the capacity of the Koopman model.

In this work, we propose a Mamba-based Koopman operator (MamKO) by integrating the matrices generation network from Mamba (Gu & Dao, 2023) with the original Koopman model to enhance the modeling accuracy. The structure can generate the Koopman operators according to the historical data and construct the SSM accordingly. Using the linear SSM, the MPC problem is formulated as a convex optimization problem. The proposed methods are then evaluated against existing representative approaches across various settings. The results demonstrate the superiority of the proposed framework in both modeling and control performance compared to state-of-the-art methods. The contributions of this work are summarized as follows:

- We propose a novel Mamba-based Koopman operator (MamKO) modeling method, which leverages matrices generated from the Mamba structure to model complex nonlinear systems.

- We further enhance the matrix generation structure of Mamba to accommodate unstable and time-varying systems.

- We develop an MPC scheme based on the MamKO model to achieve computationally efficient optimal control of nonlinear systems.

- Experiments are conducted on benchmark time-invariant and time-varying systems to illustrate the efficacy and superiority of the proposed framework.

## 2 RELATED WORKS

**Koopman Operators** Koopman operator theory (Koopman, 1931) has been regarded as a powerful tool for dynamic system analysis (Brunton et al., 2022). In (Schulze et al., 2022), the scope of Koopman operator theory is extended from autonomous systems to controlled systems, enabling the design of corresponding controllers. Methods with manually selected observable functions show promising results in both modeling and control (Zhang et al., 2023). However, the time consumption to find appropriate observable functions is a significant drawback of this approach. An alternative method using NNs to automatically construct the observable functions can bypass this problem. This kind of Koopman model has been effectively applied in areas such as fluid physics (Morton et al., 2018), robotics (Shi & Meng, 2022), chemical engineering (Han et al., 2024a), and vehicle systems (Chen & Lv, 2024), where the neural network-based model serves as a powerful predictor.

Although the Koopman model has received much attention and is well-studied both in the field of modeling and control, challenges still exist when facing complex systems, especially for time-varying systems. (Zhang et al., 2019) has developed an online dynamic mode decomposition strategy for time-varying systems with slowly changing parameters. With the online collected data, (Hao et al., 2022) designs an adjusting algorithm to tune the parameters in the observable functions. Integrating with the Fourier filter to disentangle and exploit time-invariant and time-varying dynamics, the Koopman operators are updated in (Liu et al., 2023) to accommodate the time-varying dynamics. While these methods have shown promising results, the optimization process within the online updating framework can be time-consuming and may lead to infeasible solutions. In contrast, our approach directly generates the Koopman operators from historical data, enabling real-time production of the time-varying SSM.

**Mamba Structure** Previous works on Mamba are based on the research of state-space model (Gu et al., 2020; 2022b) for continuous data processing and applied to the field of audio and vision (Goel et al., 2022; Nguyen et al., 2022). The structured state-space sequence models (S4) (Gu et al., 2022a) are expanded to selective state-space models (S6) in (Gu & Dao, 2023) by adding the selection mechanism. The mechanism inside the Mamba structure transfers the time-invariant model to the time-varying model by generating the matrices of SSMs using the input data, which can facilitate the modeling of time-varying tasks. The modeling ability has largely increased with this mechanism, and a corresponding hardware-aware algorithm has been proposed to boost computational efficiency further. The Mamba structure has been applied in the fields including language processing (Lieber et al., 2024), image segmentation (Zhu et al., 2024; Ruan & Xiang, 2024), video processing (Li et al., 2024). Notably, while the Mamba structure is based on a linear SSM, it is not directly suitable for controller design due to nonlinear functions within the selection mechanism. Modifications have been introduced to adapt the system to fit within the control framework.

**Model-based Learning Control** With the development of data-driven methods, the modeling accuracy has increased, facilitating the application of model-based learning control methods. In the field of model-based reinforcement learning, multiple algorithms have been proposed to train the control strategy inside the established model (Ha & Schmidhuber, 2018; Hafner et al., 2019; 2020; 2023). From the perspective of MPC, the computational burden for optimization problems is an important issue for model-based learning control. Some machine learning methods are preferred for lightweight use. Stochastic MPC based on Gaussian regression are discussed in (Hewing et al., 2019; 2020). Gaussian regression is applied to learn unknown disturbances inside the systems, which improves the robustness of the controller. From the collected input-output trajectories, a linear model can be constructed in the form of the Hankel matrix and helps to generate optimal control output (Berberich et al., 2020). Neural network-based models have also been integrated into the MPC framework (Chen et al., 2019; Nubert et al., 2020). However, the computational burden becomes a significant concern as the modeling accuracy improves from the intricate structures. In this work, we incorporate the Mamba structure and Koopman operator theory to balance modeling performance with control efficiency.

## 3 Modeling

In this section, the basic concepts of the Koopman operator will first be introduced. Then, the derivation of the Mamba-based Koopman operator (MamKO) is presented.

### 3.1 The Koopman operator

In typical Koopman-based approaches, a general time-invariant controlled nonlinear system is considered presented as:

$$x_{k+1} = f(x_k, u_k) \tag{1}$$

where $x_k \in \mathcal{X} \subset \mathbb{R}^n$ is the state vector at time instant $k$; $u_k \in \mathcal{U} \subset \mathbb{R}^m$ denotes the input vector of the system at time instant $k$; $f$ is a nonlinear function describing the dynamic behavior of the nonlinear system. The Koopman theory proposed in Koopman (1931) shows that there exists an infinite-dimensional Koopman operator $\mathcal{K} : \mathcal{H} \to \mathcal{H}$ acting on the observable functions such that the dynamics of the nonlinear process in (1) can be linearly represented, which can be formulated as:

$$\mathcal{K}\psi(x_k, u_k) = \psi \circ f(x_k, u_k) = \psi(x_{k+1}) \tag{2}$$

where $\psi$ denotes the observable functions and $\circ$ represents function composition. From an application point of view, it is favorable to approximate Koopman operators on a finite-dimensional function space $\overline{\mathcal{H}} \subset \mathcal{H}$. Considering the Koopman operator design in (Korda & Mezić, 2018), the system in (1) modeled by Koopman operator can be formulated as:

$$\begin{aligned} z_{k+1} &= Az_k + Bu_k \\ \hat{x}_k &= Cz_k \end{aligned} \tag{3}$$

where $z_k = \psi(x_k)$ denotes the shifted state vector in $\overline{\mathcal{H}}$ via the observable function $\psi : \mathbb{R}^n \to \mathbb{R}^N$. The corresponding finite-dimensional Koopman operator is split into $A \in \mathbb{R}^{N \times N}$ and $B \in \mathbb{R}^{N \times m}$. The details for learning the Koopman operator are included in Appendix A.

### 3.2 MAMBA-BASED KOOPMAN OPERATOR

For real-world applications, the formulation in (1) could fail in modeling systems where the future states are not only dependent on the current state and inputs but also on time. The parameters or structures of the systems can change over time. A more general form considering the influence of time can be formulated as:

$$x_{k+1} = f(x_k, u_k, k), \tag{4}$$

At a given time instant $k$, the time-varying nonlinear system in (4) can be regarded as a time-invariant system in (1). Thus, the dynamics at each time instant can be represented by the Koopman model in (3), which implies that a distinct set of matrices $\{A, B, C\}$ can capture the dynamics at a specific time instant. However, as time proceeds, the matrices $\{A, B, C\}$ for the previous instant will not be sufficient for describing the dynamics at the current instant. An effective approach to model this time-varying system involves identifying a set of matrices $A_k, B_k, C_k$ for each time instant $k$, which adapts to the changing dynamics. The corresponding linear time-varying SSM can capture the dynamics of the system for each time instant, which can be formulated as:

$$\begin{aligned} z_{k+1} &= A_k z_k + B_k u_k \\ \hat{x}_k &= C_k z_k \end{aligned} \tag{5}$$

where $z_k = \psi(x_k)$ represents the shifted state vector. Some parameters from observable function $\psi_k$ and Koopman operators $A_k$, $B_k$, and $C_k$ should be updated online to accommodate time-varying systems. Some research has been dedicated to updating parameters in $\psi_k$, $A_k$, $B_k$, and $C_k$ using online-collected datasets. In (Hao et al., 2022; Chen et al., 2024; Zhang et al., 2019), the Koopman operators or observable functions are optimized online to approximate the dynamics reflected in the newly collected data. Although these methods can provide good performance, solving the corresponding online optimization problems can be time-consuming and even infeasible. Instead, in our approach, we resort to the Mamba structure, which contains a generative framework for matrices, to enable timely real-time updates of the Koopman matrices.

**Mamba Structure** Mamba (Gu & Dao, 2023), a large language model based on the state-space model, provides a promising frame for modeling time-varying systems. Compared with the S4 framework (Gu et al., 2022a) for time-invariant systems, the selection mechanism is included to generate matrices from the input sequence, which formulated the time-varying SSM.

Inspired by the SSM model from the Mamba framework, we aim to develop a generative approach to compute the time-varying Koopman operators over the prediction horizon. However, directly applying the Mamba framework for the modeling and predictive control task is impractical. When considering the system in (5), the Mamba framework only focuses on the inputs $u$ in the form of the word sequence, while states in the LLMs do not have exact meaning. However, for a controlled system in (4), the states and inputs with physical meaning should all be considered. Meanwhile, in (5), as $B_k$ is generated from the input sequence containing $u_k$, the bilinear term $B_k u_k$ can lead to a non-convex optimization problem for in the control part. As a substitute, we generate the matrices from the historical data. The differences between the proposed framework and the Mamba framework are discussed in Appendix B.

**Matrices Generation** Inspired by the Mamba framework, we utilize discretization to generate matrices efficiently. Given the matrices $A$, $B$ in the continuous system and the sampling period $T$, the discrete matrices $\bar{A}$, $\bar{B}$ can be calculated by the zero-order hold (ZOH) as:

$$\begin{aligned} \bar{A} &= e^{AT} \\ \bar{B} &= \int_0^T e^{At} \mathrm{d}t \cdot B \end{aligned} \tag{6}$$

The matrix $A$ is set as a diagonal matrix, facilitating the discretization process. In the Mamba framework, the diagonal elements of $A$ are constrained to be negative through a negative exponential function to promote the stability of the SSM. However, not all real-world systems are inherently stable, such as the oscillator system (Elowitz & Leibler, 2000). This constraint can significantly compromise modeling accuracy. In our work, we address this problem by substituting the negative exponential function with a negative continuously differentiable exponential linear unit (CELU) (Barron, 2017). The formulation of CELU can be presented as:

$$CELU(x) = \max(0, x) + \min(0, \alpha(e^{(\frac{x}{\alpha})} - 1)) \tag{7}$$

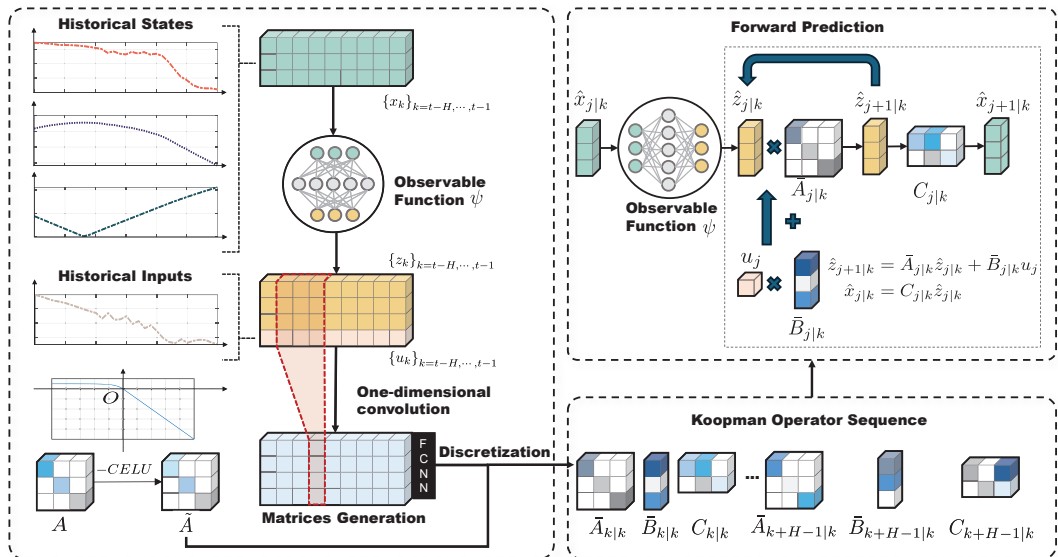

Figure 1: Structure of the MamKO. With the matrices generation network and the observable function, the linear time-varying SSM can be established. By transforming the initial state to the shifted state through the observable function $\psi$, the linear time-varying SSM can produce a sequence of the future states.

where $\alpha = 1$ in this paper. The CELU is continuously differentiable, which can lead to smooth gradient updates. As CELU can generate negative output, the negative CELU allows for the existence of positive eigenvalues, thereby facilitating the representation of unstable systems. The new diagonal matrix activated by CELU is denoted as $\tilde{A}$, with the eigenvalues limited in the range of $[-\infty, 1]$.

At time instant $k$, the historical state sequence $x_{k-H:k-1}$ from the past $H$ steps is shifted to space $\overline{\mathcal{H}}$ through the observable function $\psi$, obtaining the shifted state sequence $z_{k-H:k-1}$. The combination of shifted state sequence and historical input sequence forms the historical data sequence $\left[z_{k-H:k-1}^{\mathrm{T}}, u_{k-H:k-1}^{\mathrm{T}}\right]^{\mathrm{T}}$. One-dimensional convolution is implemented in the historical data sequence to extract the temporal information inside. The extracted feature $c_i$ can be calculated by:

$$c_i = w \left[z_{i:i+h-1}^{\mathrm{T}}, u_{i:i+h-1}^{\mathrm{T}}\right]^{\mathrm{T}} + b, i = k - H, \ldots, k - h \qquad (8)$$

where $w$ and $b$ are the weight and bias parameters in the convolution network, and $h$ is the kernel size. The matrix sequences $B_{k:k+H-1}$ and $C_{k:k+H-1}$ and the sampling periods sequence $T_{k:k+H-1}$ are encoded from the extracted features using fully connected NNs (FCNNs). Discretizing $\tilde{A}$, $B_{k:k+H-1}$ using sampling periods sequence $T_{k:k+H-1}$ by (6), the obtained matrix sequences $\bar{A}_{k:k+H-1}$, $\bar{B}_{k:k+H-1}$ and $C_{k:k+H-1}$ are applied to the Koopman framework to accomplish multi-step-ahead prediction tasks via the model in (5) recursively. The structure of the proposed MamKO is presented in Figure 1.

**Optimization Procedure** The observable function that projects the state vector to finite-dimensional function space is denoted as $\psi(\cdot|\theta_\psi)$ with trainable parameters $\theta_\psi$. The matrices generation networks can be denoted as $\phi(\cdot|\theta_\phi)$ with trainable parameters $\theta_\phi$. The networks take the data sequence $\left[z_{k-H:k-1}^{\mathrm{T}}, u_{k-H:k-1}^{\mathrm{T}}\right]^{\mathrm{T}}$ and trainable matrix $A$ as input, and produce matrix sequence $\bar{A}_{k:k+H-1|k}$, $\bar{B}_{k:k+H-1|k}$ and $C_{k:k+H-1|k}$. To effectively train the matrix $A$, parameters $\theta_\psi$ and $\theta_\phi$, the optimization problem for multi-step-ahead prediction tasks should be designed. Given the dataset

$\{x_j, u_j\}_{j=k-H,\cdots,k+H}$, the optimization can be formulated as follows:

$$\min_{A,\theta_\psi,\theta_\phi} \frac{1}{H} \sum_{j=k+1}^{k+H} \|\hat{x}_{j|k} - x_j\|_2^2 \tag{9a}$$

$$\text{s.t. } \hat{z}_{j+1|k} = \bar{A}_{j|k}\hat{z}_{j|k} + \bar{B}_{j|k}u_j \tag{9b}$$

$$\hat{z}_{k|k} = \psi(x_k|\theta_\psi) \tag{9c}$$

$$\hat{x}_{j|k} = C_{j|k}\hat{z}_{j|k} \tag{9d}$$

$$\tilde{A} = -CELU(A) \tag{9e}$$

$$z_j = \psi(x_j|\theta_\psi), j = k - H, \cdots, k - 1 \tag{9f}$$

$$\bar{A}_{k:k+H-1|k}, \bar{B}_{k:k+H-1|k}, C_{k:k+H-1|k} = \phi(\tilde{A}, z_{k-H:k-1}, u_{k-H:k-1}|\theta_\phi) \tag{9g}$$

where $\hat{z}_{j|k}$ represents the predicted shifted state vector for time instant $j$ at time instant $k$; $\hat{x}_{j|k}$ represents the state vector for time instant $j$ predicted at time instant $k$; $H$ represents the length of the model's prediction horizon. $\bar{A}_{k:k+H-1|k}, \bar{B}_{k:k+H-1|k}$ and $C_{k:k+H-1|k}$ are the generated matrix sequences from the historical data sequence. To minimize the prediction loss in (9a), all parameters are updated with gradient descent using ADAM (Kingma, 2014).

## 4 CONTROL

In this section, for the time-varying system in (4), we propose a model predictive control framework that is inspired by Korda & Mezić (2018) based on the MamKO model.

At each time instant $k$, the matrix sequences $\bar{A}_{k:k+H-1|k}, \bar{B}_{k:k+H-1|k}$ and $C_{k:k+H-1|k}$ for prediction are generated previously with the historical data. With the matrix sequence, the optimization problem for MPC is formulated as follows:

$$\min_{u_{k|k}^*,\ldots,u_{k+H-1|k}^*} \sum_{j=k+1}^{k+H-1} \left( \|C_{j|k}\hat{z}_{j|k} - x_s\|_Q^2 + \|u_{j|k} - u_{j-1|k}\|_R^2 \right) + \|C_{k+H|k}\hat{z}_{k+H|k} - x_s\|_P^2 \tag{10a}$$

$$\text{s.t. } z_{j+1|k} = \bar{A}_{j|k}\hat{z}_{j|k} + \bar{B}_{j|k}u_{j|k} \tag{10b}$$

$$\hat{z}_{k|k} = \psi(x_k) \tag{10c}$$

$$C\hat{z}_{j|k} \in \mathcal{X} \tag{10d}$$

$$u_{j|k} \in \mathcal{U}, j = k + 1, \ldots, k + H - 1 \tag{10e}$$

where $u_{k|k}^*, \ldots, u_{k+N-1|k}^*$ denotes the trajectory of optimal control signals determined given the current time instant $k$; $\|\cdot\|_Q^2$ denotes a vector's squared weighted Euclidean norm, which is computed by $\|x\|_Q^2 = x^TQx$; $Q$, $R$ and $P$ are positive-definite weighting matrices; $\|C_{j|k}\hat{z}_{j|k} - x_s\|_Q^2$ and $\|C_{k+H|k}\hat{z}_{k+H|k} - x_s\|_P^2$ are used to reduce tracking errors; $\|u_{j|k} - u_{j-1|k}\|_R^2$ is set to improve the smoothness of the control inputs; (10b) serves as the model constraint; (10c) provides the initial condition in the shifted state-space; (10d) and (10e) are state and input constraints, respectively. At time instant $k \geq 0$, the optimal control input trajectory is calculated and the first element of the trajectory, $u_{k|k}^*$ is applied to the nonlinear process in (4) as the control input for the current time instant. The convex optimization problems derived from linear SSMs can be efficiently solved.

## 5 EXPERIMENTS

In this section, we will evaluate the performance of the MamKO model in terms of modeling and control. Specifically, we evaluate the following aspects: (a) **Convergence** of the proposed training algorithm on different systems with random parameter initialization. (b) **Model accuracy** of the MamKO compared to other methods in time-invariant and time-varying systems. (c) **Control performance** of the MPC based on the MamKO compared to other methods in time-invariant and time-varying systems.

Five benchmark systems are included to evaluate the modeling and control performance of the MamKO. The CartPole balancing system is included as a benchmark system for the controller design task, which has been widely used in deep reinforcement learning (DRL) research (Lillicrap, 2015; Haarnoja et al., 2018). Correspondingly, a time-varying CartPole balancing system described in (Hao et al., 2022) is considered to test the performance of MamKO on the time-varying system. Moreover, we apply the MamKO to the biological gene regulatory networks (GRN) (Elowitz & Leibler, 2000). The complex nonlinear systems in the chemical engineering field are also considered. The reactor-separator chemical process (Yin & Liu, 2019) comprising two continuous stirred tank reactors and a flash tank separator is chosen as a benchmark chemical process, denoted as the reactor-separator chemical process (RSCP) system. We also simulate the reactor-separator process with time-varying parameters (Nikravesh et al., 2000), denoted as the time-varying RSCP system.

For the modeling performance evaluation, we compare the proposed method with two competitive baseline methods. **(1)** The Deep Koopman Operator (DKO) (Lusch et al., 2018; Han et al., 2020) is a Koopman method that employs deep NNs in constructing the observable functions. Different from the generative Koopman operators from MamKO, the Koopman operators in the DKO are invariant. **(2)** The Multilayer Perceptrons (MLP) (Chua et al., 2018), based on multilayer fully connected NNs, is implemented as a baseline for modeling the systems.

For the control performance evaluation, we compare the MamKO-based MPC with three state-of-the-art baseline methods: **(1)** The DKO-based MPC (Lusch et al., 2018; Han et al., 2020); **(2)** MLP-based MPC, which utilizes the NNs as predictive models and solves a nonlinear optimization with the help of Casadi (Andersson et al., 2019) and Interior Point Optimizer, pronounced eye-pea-Opt (IPOPT) (Wächter & Biegler, 2006); **(3)** Soft actor-critic (SAC) (Haarnoja et al., 2018), which is a state-of-the-art model-free reinforcement learning algorithm. Despite the higher sample complexity of model-free methods compared to model-based ones, SAC often achieves superior control performance. SAC updates the controller to minimize cumulative stage costs, thereby implicitly optimizing for a stabilizing controller.

For each environment, trajectories of state and action samples are gathered, generating a training set of $36,000$ samples, a validation set of $4,000$ samples, and a test set of $4,000$ samples. Specifically, actions are uniformly sampled from the action space for each time instant for the CartPole system, the time-varying CartPole system, and the GRN system. For the RSCP and time-varying RSCP systems, actions are determined using a step function with added random noise. Details of the experimental setup are provided in Appendix C. The methods are trained to predict state sequences over a 30-step horizon. Hyperparameters for each method are listed in Appendix D. For the SAC control task, each environment undergoes training with 1000k steps of state-action-reward pairs.

## 5.1 Modeling evaluation

The losses for the modeling task are calculated by the average prediction error of each step on the test set. For each system, we run ten model training trials with different datasets and randomly initialized parameters to test the convergence of the algorithms. The results for each system are presented in Figure 2 (a-e). In comparison to DKO, MamKO demonstrates superior modeling accuracy across all environments. Especially for the two time-varying systems, the MamKO outperforms the other two methods. Notably, with the nonlinear structures from the NNs, the MLP has fewer prediction errors in GRN and RSCP. Nonetheless, the inherent nonlinearities of the MLP model may complicate optimization tasks, leading to reduced computational efficiency and suboptimal behavior. The experiments in the next section will show the possible limitations of MLP in controller design.

**Performance on the rapidly changing dynamics** For the time-varying CartPole system in (Hao et al., 2022), a time-varying coefficient of the friction of the cart is added to the system in the form of sine waves. Based on the example, we compare the modeling performance of the three methods with different frequencies of the sine waves. The time-varying Cartpole system described before has the angular frequency of 1 rad/s. Two time-varying Cartpole systems with the angular frequencies of 0.1 rad/s and 10 rad/s are added. The results are shown in Figure 2 (f), illustrating that as the angular frequency of the time-varying parameters increases, the advantages of using MamKO become more pronounced. The generative matrices provide a more accurate representation of the system with rapidly changing parameters, as different matrices approximate the dynamics of systems at each time instant.

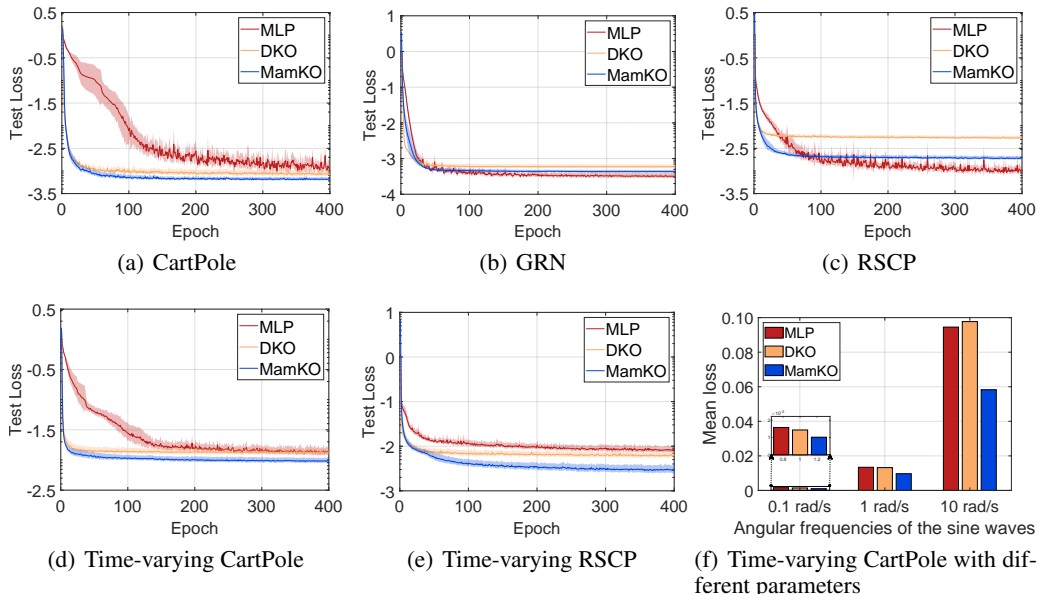

(a) CartPole      (b) GRN      (c) RSCP

(d) Time-varying CartPole      (e) Time-varying RSCP      (f) Time-varying CartPole with different parameters

Figure 2: Average prediction error on the test set (a-e) and average prediction error for the time-varying CartPole with different coefficients (f). In (a-e), the Y-axis indicates the average mean-squared prediction error in log space for the 30-step-ahead prediction task, and the X-axis indicates the training epochs. The shaded area represents the confidence interval (0.5 times the standard deviation) across ten training trials. In (f), the Y-axis indicates the average loss after training for 400 epochs.

## 5.2 CONTROL EVALUATION

The control performance of the four methods is evaluated and compared in the five systems. The time-varying CartPole with the angular frequency of 10 rad/s is included to evaluate the control performance for the system with rapidly changing dynamics. The control task for each environment is set-point tracking, and the details can be found in Appendix C. The weighting parameters $Q$, $R$, and $P$ for the MPCs are carefully adjusted to reach the best performance of each baseline method. The prediction horizon is set as 30 for all MPCs.

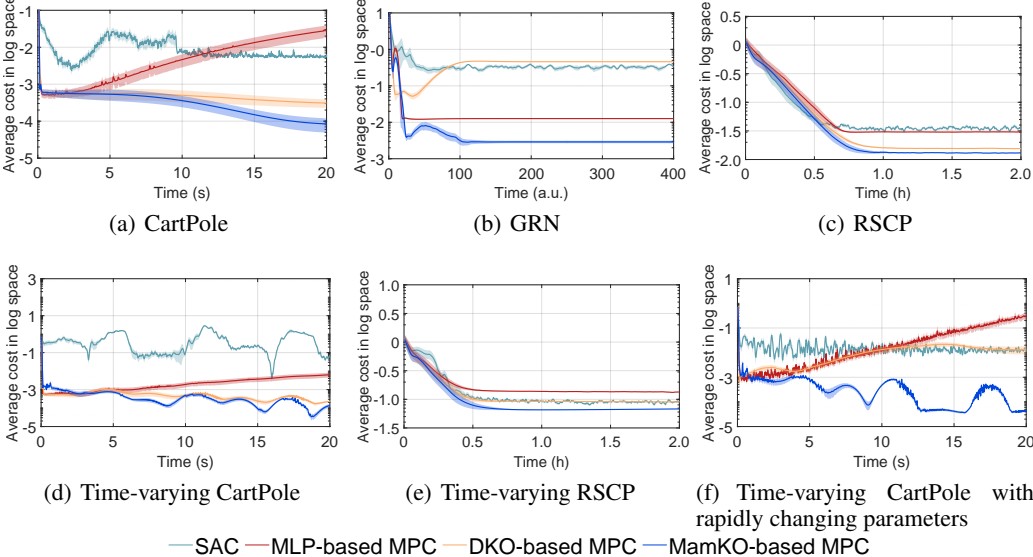

(a) CartPole      (b) GRN      (c) RSCP

(d) Time-varying CartPole      (e) Time-varying RSCP      (f) Time-varying CartPole with rapidly changing parameters

Figure 3: Average cost based on track error. The Y-axis indicates the average cost in log space for ten experiments with random initial states. The shaded area represents the confidence interval (0.3 times the standard deviation) over the ten experiments.

The cost trajectories are shown in Figure 3. The state trajectories can be found in Appendix F. As shown in Figure 3, the MamKO-based MPC achieves the best control performance for all five systems. In comparison with DKO-based MPC, the MamKO-based MPC reduces the cost of the five systems by 5.05%, 3.70%, 92.10%, 6.56%, 14.19%, 84.74%, respectively. A significant improvement is observed in the time-varying CartPole with rapidly changing parameters. Although MLP achieves better modeling performance on GRN and RSCP systems, the corresponding control performance is not as good as its modeling performance, which can be attributed to suboptimal solutions from the non-convex optimization problems. Compared with the SAC, the two Koopman model-based MPCs can all achieve a relatively stable control performance for the time-varying system. Although it can be hard to construct a time-varying model, the Koopman operator in DKO can construct an average Koopman operator that accommodates various time-varying parameters in the systems, which can provide relatively good results.

## 5.3 EVALUATION ON THE ACTIVATION FUNCTION

As we simplify $A$ as a diagonal matrix, the eigenvalues of the matrix become the trainable parameters directly. Compared to the Mamba framework, which sets the eigenvalues to be negative, we replace the original negative exponential function with the negative CELU function. In this section, we will compare the modeling performance of MamKO using the negative exponential function, the negative CELU function, and no activation function on the CartPole system, the GRN system, and the RSCP system. As observed in Figure 4, the MamKO with CELU function has the best modeling performance.

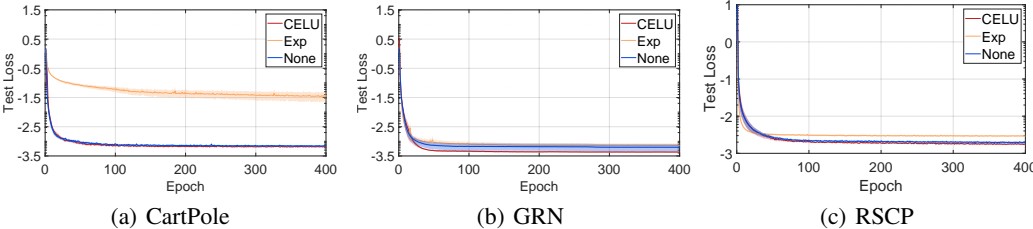

(a) CartPole        (b) GRN        (c) RSCP

Figure 4: Evaluation on the activation function for the eigenvalues. For the CartPole system, the MamKO with the negative CELU function performs slightly better than the one without the activation function and largely better than the one with the negative exponential function. For the GRN and RSCP systems, the MamKO with the negative CELU has the smallest mean loss and lowest variance. Compared with the MamKO with no activation function, the improvement of the MamKO with the negative CELU is not significant. However, the initiation of $A$ of the MamKO with no activation function should be carefully adjusted since no constraint is added to the eigenvalues. Otherwise, too large eigenvalues may result in unstable gradient descent.

## 5.4 ABLATION ON THE DISCRETIZATION

We also evaluate the impact of discretization on modeling performance. The benefit of generating the sequence of the Koopman operator through the method in (6) is demonstrated in this subsection. We compare the performance of this modeling approach with another method that applies the same discretization technique to matrix $B$ as to matrix $A$, which is $\bar{B} = e^{BT}$. The pipeline is named as $Multiplication$. The modeling results for each environment are presented in Table 1. From Table 1, we can find that with the discretization on the matrices following (6), the modeling accuracy can be significantly improved, which validates the effect of the discretization.

Table 1: Results of the ablation on the discretization.

| Method | CartPole | GRN | RSCP | Time-varying CartPole | Time-varying RSCP |
|---|---|---|---|---|---|
| Discretization | $6.50 \times 10^{-4}$ | $1.91 \times 10^{-3}$ | $2.92 \times 10^{-3}$ | $4.31 \times 10^{-4}$ | $9.67 \times 10^{-3}$ |
| Multiplication | $8.30 \times 10^{-4}$ | $3.52 \times 10^{-3}$ | $7.60 \times 10^{-3}$ | $9.11 \times 10^{-4}$ | $1.19 \times 10^{-2}$ |

## 5.5 EVALUATION ON THE COMPUTATION TIME

In this subsection, we evaluate the computation time of the control methods on different benchmark systems. The online control implementation of the control methods is conducted on a computer equipped with an Intel Core i9-13900K CPU and 128 GB DDR4 RAM. The computation times of the MamKO-based MPC and baselines during control on the benchmark systems are presented in Table 2. Compared with the MLP-based MPC, the MamKO-based MPC reduces the computation time by 98.38%, 83.39%, 99.21%, 90.34%, 99.17% for the five benchmark systems, respectively. To further demonstrate the computational efficiency of the proposed method, the sampling periods of the systems are presented in Table 3. A comparison of the sampling periods and computation times demonstrates that the proposed MamKO-based MPC method can reliably ensure online implementation for each of the considered systems. The efficient online implementation of the proposed method stems from the use of a linear state-space model within the proposed framework, which facilitates the formulation of an optimal control problem that requires solving convex optimization despite the nonlinearity in the dynamics of the considered systems.

Table 2: Results of the computational time of each method in the benchmark systems.

| Method | CartPole | GRN | RSCP | Time-varying CartPole | Time-varying RSCP |
|---|---|---|---|---|---|
| MLP-based MPC | $7.43 \times 10^{-1}$ s | $1.09 \times 10^{-1}$ s | 3.31 s | $1.74 \times 10^{-1}$ s | 3.45 s |
| DKO-based MPC | $7.35 \times 10^{-3}$ s | $1.41 \times 10^{-2}$ s | $1.06 \times 10^{-2}$ s | $7.63 \times 10^{-3}$ s | $1.28 \times 10^{-2}$ s |
| MamKO-based MPC | $1.02 \times 10^{-2}$ s | $1.81 \times 10^{-2}$ s | $2.62 \times 10^{-2}$ s | $1.68 \times 10^{-2}$ s | $2.95 \times 10^{-2}$ s |
| SAC | $2.67 \times 10^{-4}$ s | $3.23 \times 10^{-4}$ s | $3.41 \times 10^{-4}$ s | $2.68 \times 10^{-4}$ s | $3.32 \times 10^{-4}$ s |

Table 3: Sampling periods of the benchmark systems.

| Method | CartPole | GRN | RSCP | Time-varying CartPole | Time-varying RSCP |
|---|---|---|---|---|---|
| Sampling period | 0.02 s | 1 s | 18 s | 0.02 s | 18 s |

The results also indicate that the extended computation time required by the MLP-based MPC, which is due to the need to solve nonlinear optimization problems, poses challenges for its online implementation. This limitation is particularly critical for systems with fast dynamics and short sampling periods, and for larger-scale systems with numerous control inputs and state variables to optimize.

## 6 CONCLUDING REMARKS

In this paper, a new modeling and control framework called Mamba-based Koopman Operator (MamKO) was proposed by seamlessly integrating the Koopman operator with the large language model Mamba. The matrices generation network from Mamba was adapted to construct the linear time-varying state-space model based on the Koopman modeling concept. The MamKO model can effectively predict the future states of nonlinear systems with time-varying parameters. A model predictive controller was formulated based on the established MamKO model. Both time-invariant and time-varying benchmark systems were leveraged to evaluate the modeling and control performance of the proposed method. The experiments demonstrated the superior performance of the MamKO model in both modeling and control tasks. Future work will focus on applying the MamKO model to large-scale systems to further evaluate its modeling capabilities.

Future research directions include the formal analysis of the convergence and stability of the proposed MamKO-based control approach and the utilization of the MamKO-based modeling framework for energy optimization of energy-intensive industrial systems, for example, through integrating the proposed modeling framework with Koopman-based economic model predictive control (Han et al., 2024b). We also plan to investigate the application of the MamKO-based control methods to industrial systems of larger scales and more complex dynamics for efficient and robust system operations.

ACKNOWLEDGMENTS

This research is supported by the National Research Foundation, Singapore, and PUB, Singapore's National Water Agency under its RIE2025 Urban Solutions and Sustainability (USS) (Water) Centre of Excellence (CoE) Programme, awarded to Nanyang Environment & Water Research Institute (NEWRI), Nanyang Technological University, Singapore (NTU). This research is also supported by Ministry of Education, Singapore, under its Academic Research Fund Tier 1 (RS15/21 & RG63/22). Any opinions, findings and conclusions or recommendations expressed in this material are those of the author(s) and do not reflect the views of the National Research Foundation, Singapore, and PUB, Singapore's National Water Agency.

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

## A    KOOPMAN OPERATOR

According to the approach discussed in (Korda & Mezić, 2018), a state vector incorporating both state and input can be defined as $\mathcal{X}_k = \left[x_k^{\mathrm{T}}, u_k^{\mathrm{T}}\right]^{\mathrm{T}}$. Defining a left shift operator $\mathcal{S}$ such that $\mathcal{S}u_k = u_{k+1}$, the dynamics of this augmented state vector can be expressed as follows:

$$\mathcal{X}_{k+1} = \mathcal{F}(\mathcal{X}_k) := \begin{bmatrix} f(x_k, u_k) \\ \mathcal{S}u_k \end{bmatrix} := \begin{bmatrix} f(x_k, u_k) \\ u_{k+1} \end{bmatrix} \tag{11}$$

The Koopman operator $\mathcal{K}_{\mathcal{F}} : \mathcal{H}_{\mathcal{F}} \to \mathcal{H}_{\mathcal{F}}$ for (11) governs the dynamics of the augmented vector $\mathcal{X}$ in a linear fashion:

$$\mathcal{K}_{\mathcal{F}}\Psi(\mathcal{X}_k) = \Psi \circ \mathcal{F}(\mathcal{X}_k) = \Psi(\mathcal{X}_{k+1}). \tag{12}$$

The observable functions inside $\Psi$ transfer the original states into the higher-dimensional space.

In (Korda & Mezić, 2018), only the states are lifted to a higher-dimensional space, while no transformations are applied to the inputs. Specifically, the observable functions $\Psi$ are defined as follows:

$$\Psi(\mathcal{X}_k) = \Psi(x_k, u_k) = \left[\psi^{\mathrm{T}}(x_k), u_k^{\mathrm{T}}\right]^{\mathrm{T}} \tag{13}$$

where $\psi : \mathbb{R}^n \to \mathbb{R}^N$ is the observable function that transforms the original states.

For real-world applications, a finite-dimensional approximation of the Koopman operator, denoted as $\mathcal{K}_{N_\phi}$, needs to be identified. Since future inputs are not considered, only the elements in the first $N$ rows of the $\mathcal{K}_{N_\phi}$ need to be identified. The formulation of $\mathcal{K}_{N_\phi}$ can be expressed as:

$$\mathcal{K}_{N_\phi} = \left[ \begin{array}{c|c} A & B \\ \hline * & * \end{array} \right] \tag{14}$$

where $A \in \mathbb{R}^{N \times N}$, $B \in \mathbb{R}^{N \times m}$. To capture the dynamics of $\mathcal{X}_k$, it is sufficient to identify only the matrices $A$ and $B$ in (14), while the blocks denoted by $*$ can be ignored.

## B    MAMBA FRAMEWORK

The matrices generation network in MamKO is inspired by the Mamba. It is worth noting that there are differences between our methods and the Mamba structure.

Firstly, the Mamba structure is designed for LLM with word sequences as inputs and generates corresponding word sequences. Instead, our method attempts to model real-world systems containing both states and inputs. Compared with LLM, our MamKO model is closer to the concept (Ha & Schmidhuber, 2018), which contains the interaction between states and actions.

Secondly, the matrices generation network of Mamba utilizes the information in the whole input sequence. Given the sequence $U_k = \{u_j\}_{j=k,\cdots,k+H-1}$, the matrices $A_k$, $B_k$ and $C_k$ in (5) are obtained from the nonlinear neural networks, which can be denoted as $A_k = f_{A_k}(U_k)$, $B_k = f_{B_k}(U_k)$. $C_k = f_{C_k}(U_k)$. The SSM from Mamba can be formulated as:

$$\begin{aligned} z_{k+1} &= f_{A_k}(U_k)z_k + f_{B_k}(U_k)u_k \\ \hat{x}_k &= f_{C_k}(U_k)z_k \end{aligned} \tag{15}$$

The nonlinear functions $f_{A_k}$, $f_{B_k}$, $f_{C_k}$ and the bilinear terms $f_{A_k}(U_k)z_k$, $f_{B_k}(U_k)u_k$ make the relationship between $z_k$ and $u_k$ no longer linear. Applying MPC for this nonlinear system can cause non-convex optimization problems, which may reduce the control efficiency. In our work, the matrices are generated from historical data, which have no direct relationship between $z_k$ and $u_k$. The linear relationship between $z_k$ and $c_k$ allows for the construction of convex optimization problems.

Lastly, the SSMs in the Mamba structure are separately generated for each channel of the input sequence. For the input $u \in \mathbb{R}^m$ with $m$ channels, $m$ pairs of SSMs with $A_k \in \mathbb{R}^{N \times N}$, $B_k \in \mathbb{R}^{N \times 1}$, $C_k \in \mathbb{R}^{1 \times N}$ matrices are generated at time instant $k$. The multiple pairs of SSMs are not directly applicable to MPC. In our framework, the SSMs are designed to describe the dynamics of all the input channels (features), which can be formulated as $A_k \in \mathbb{R}^{N \times N}$, $B_k \in \mathbb{R}^{N \times m}$, $C_k \in \mathbb{R}^{n \times N}$ for each time instant.

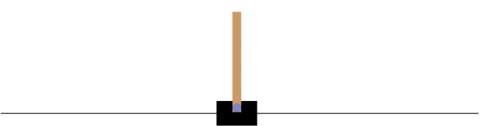

Figure 5: CartPole system.

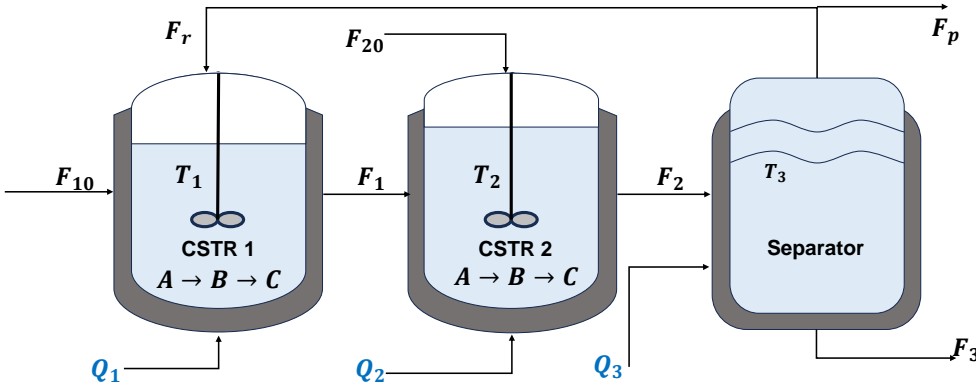

Figure 6: Time-invariant RSCP system (Liu et al., 2008).

## C    EXPERIMENTAL SETUP

The experiments are set based on OpenAI Gym (Brockman et al., 2016). Part of the environments are presented in Figure 5 and Figure 6.

### C.1    CARTPOLE - INVERTED PENDULUM ON A CART

We modified the CartPole system described in (Brockman et al., 2016) using a continuous action space instead of a discrete one. The system consists of a cart that moves horizontally with an inverted pendulum attached to it. While the cart is fully actuated, the pendulum remains unactuated. In this experiment, the controller aims to keep the pendulum upright and vertical. The control input is the horizontal force applied to the cart ($a \in [-20, 20]$). $x_{\text{threshold}}$ and $\theta_{\text{threshold}}$ represents the maximum of position and angle, respectively, $x_{\text{threshold}} = 10$ and $\theta_{\text{threshold}} = 20°$. The episode ends if $|\theta| > \theta_{\text{threshold}}$ and the episodes end in advance. The episodes for control evaluation are of length 1000. For time-varying CartPole, the model is based on a modified system in (Hao et al., 2022), which can be presented as:

$$\ddot{x}_t = \frac{F_t + ml\left(\dot{\bar{\theta}}_t^2 \sin \bar{\theta}_t - \ddot{\bar{\theta}}_t \cos \bar{\theta}_t\right) - \mu_t^c \operatorname{sgn}(\dot{x}_t)}{m_c + m}$$
$$\ddot{\bar{\theta}}_t = \frac{\cos \theta_t \left[-F_t - ml\dot{\bar{\theta}}_t^2 \sin \theta_t + \mu_t^c \operatorname{sgn}(\dot{x}_t)\right] / (m_c + m)}{l\left[\frac{4}{3} - \left(m \cos^2 \bar{\theta}_t\right) / (m_c + m)\right]} + \frac{g \sin \theta_t - \mu_p \dot{\bar{\theta}}/ml}{l\left[\frac{4}{3} - \left(m \cos^2 \bar{\theta}_t\right) / (m_c + m)\right]} \tag{16}$$

where $\mu_t^c = 0.0005 + cos(\omega t)$ is time-varying coefficient of friction of cart on the track; $\omega$ is the angular frequency of the sine wave; $x_t$ is the distance to the initial position; $\bar{\theta}_t$ is the offset angle; $\dot{x}_t, \ddot{\bar{\theta}}_t$ is the velocity and the angular velocity respectively; $F_t$ is the continuous control input. The parameters are set the same as the time-invariant CartPole, which can be presented as:

$$m_c = 1.0 \, \text{kg}, m = 0.1 \, \text{kg}, l = 0.5 \, \text{m}$$

For the time-varying Cartpole system, the angular frequency is set as $\omega = 1$; for the rapidly changing dynamics, the angular frequency is set as $\omega = 0.1, \omega = 1, \omega = 10$, respectively.

Notably, the control tasks for the two systems include position control and angular control. The controller should stabilize the CartPole in the state that $x_t = 0$ and $\theta_t = 0$. The cost designed for the two CartPole systems is defined as:

$$cost_{cartpole}(x_t, \theta_t) = \frac{x_t^2}{100} + 180\frac{\theta_t^2}{\pi} \tag{17}$$

## C.2 SYNTHETIC BIOLOGY GENE REGULATORY NETWORKS

The gene regulatory networks (GRNs) discussed in this paper function at the nanoscale, demonstrating unique physical properties that distinguish them from other examples. Notably, these GRNs exhibit intriguing oscillatory behavior.

In this example, we focus on a classical dynamical system from systems and synthetic biology, specifically addressing a reference tracking task. The GRN under consideration is a synthetic three-gene regulatory network, where the dynamics of mRNAs and proteins inside the network exhibit oscillatory behavior. A detailed description of the network's mechanisms can be found in (Elowitz & Leibler, 2000). The discrete-time mathematical model of the GRNs can be represented by the following equations:

$$
\begin{aligned}
x_1(t+1) &= x_1(t) + dt \cdot \left[ -\gamma_1 x_1(t) + \frac{a_1}{K_1 + x_6^2(t)} + u_1 \right] + \xi_1(t), \\
x_2(t+1) &= x_2(t) + dt \cdot \left[ -\gamma_2 x_2(t) + \frac{a_2}{K_2 + x_4^2(t)} + u_2 \right] + \xi_2(t), \\
x_3(t+1) &= x_3(t) + dt \cdot \left[ -\gamma_3 x_3(t) + \frac{a_3}{K_3 + x_5^2(t)} + u_3 \right] + \xi_3(t), \\
x_4(t+1) &= x_4(t) + dt \cdot \left[ -c_1 x_4(t) + \beta_1 x_1(t) \right] + \xi_4(t), \\
x_5(t+1) &= x_5(t) + dt \cdot \left[ -c_2 x_5(k) + \beta_2 x_2(t) \right] + \xi_5(t), \\
x_6(t+1) &= x_6(t) + dt \cdot \left[ -c_3 x_6(t) + \beta_3 x_3(t) \right] + \xi_6(t).
\end{aligned}
\tag{18}
$$

In this context, $x_1, x_2, x_3$ represent the concentrations of the mRNA transcripts, and $x_4, x_5, x_6$ represent the concentrations of the corresponding proteins for genes 1, 2, and 3, respectively. The terms $\xi_i$, for all $i$, are independent and identically distributed. Uniform noise variables are drawn from the range $[-\delta, \delta]$. For the simulations described in Section 5, $\delta$ is set to 0, while for the robustness tracking task in F.6, $\delta$ is set to 0.5. The parameters $a_1$, $a_2$, and $a_3$ indicate the highest promoter strengths for the respective genes. The symbols $\gamma_1$, $\gamma_2$, and $\gamma_3$ represent the degradation rates of mRNA, whereas $c_1$, $c_2$, and $c_3$ correspond to the degradation rates of proteins. Moreover, $\beta_1$, $\beta_2$, and $\beta_3$ denote the protein synthesis rates, and $K_1$, $K_2$, and $K_3$ are the associated dissociation constants. The equations in (18) present a network structure in which gene 1 is inhibited by gene 2, gene 2 is suppressed by gene 3, and gene 3, in turn, is repressed by gene 1. The discretization time step is denoted by $dt$. The protein concentrations are observable and are typically measured using fluorescent markers, such as green fluorescent protein (GFP) or red fluorescent protein (mCherry). The control inputs $u_i$ are applied via light control signals, which activate photo-sensitive promoters to induce gene expression. In line with the standard practice of GRN models in (Elowitz & Leibler, 2000), we simplify the system dynamics by assuming that the parameters regulating mRNA and protein dynamics are uniform across all genes. Further details on the mathematical modeling and control of synthetic gene regulatory networks can be found in (Strelkowa & Barahona, 2010; Sootla et al., 2013). In the simulations mentioned in this paper, the parameters are chosen as follows:

$$\forall i : \ K_i = 1, a_i = 1.6, \gamma_i = 0.16, \beta_i = 0.16, c_i = 0.06, dt = 1$$

For the tracking task for $x_4$, the cost function for the GRN system is defined as:

$$cost_{GRN}(x_4) = (x_4 - 6)^2 \tag{19}$$

## C.3 REACTOR-SEPARATOR PROCESS

A reactor-separator process, consisting of two continuous stirred tank reactors (CSTRs) and a flash tank separator (Liu et al., 2008), is considered in this paper. A schematic of the process is shown in

Figure 6. The process consists of two chemical reactions: firstly, the reactant $A$ is transformed into the desired product $B$, followed by a secondary reaction where a fraction of $B$ is further converted into the byproduct $C$. The system starts with the pure reactant $A$ being fed into the first reactor (CSTR 1) at a flow rate of $F_{10}$. The output stream from CSTR 1, with a flow rate of $F_1$, together with an additional fresh supply of pure $A$ at a rate of $F_{20}$, enters CSTR 2. The outflow from CSTR 2 is then directed to the separator at a flow rate of $F_2$. Within the separator, a recycle stream with a flow rate of $F_r$ is redirected back to the first reactor for further conversion. All three vessels (CSTR 1, CSTR 2, and the separator) are equipped with a jacket to provide or remove heat, with a heating input rate $Q_i$, where $i = 1, 2, 3$. The system's ordinary differential equations can be expressed as follows (Liu et al., 2008; Yin & Liu, 2019):

$$\frac{dx_{A1}}{dt} = \frac{F_{10}}{V_1}(x_{A10} - x_{A1}) + \frac{F_r}{V_1}(x_{Ar} - x_{A1}) - \phi_c k_1 e^{\frac{-E_1}{rT_1}} x_{A1}$$

$$\frac{dx_{B1}}{dt} = \frac{F_{10}}{V_1}(x_{B10} - x_{B1}) + \frac{F_r}{V_1}(x_{Br} - x_{B1}) + \phi_c k_1 e^{\frac{-E_1}{rT_1}} x_{A1} - \phi_c k_2 e^{\frac{-E_2}{rT_1}} x_{B1}$$

$$\frac{dT_1}{dt} = \frac{F_{10}}{V_1}(T_{10} - T_1) + \frac{F_r}{V_1}(T_3 - T_1) - \phi_c \frac{\Delta H_1}{c_p} k_1 e^{\frac{-E_1}{rT_1}} x_{A1} - \phi_c \frac{\Delta H_2}{c_p} k_2 e^{\frac{-E_2}{rT_1}} x_{B1} + \frac{Q_1}{\rho c_p V_1}$$

$$\frac{dx_{A2}}{dt} = \frac{F_1}{V_2}(x_{A1} - x_{A2}) + \frac{F_{20}}{V_2}(x_{A20} - x_{A2}) - \phi_c k_1 e^{\frac{-E_1}{rT_2}} x_{A2}$$

$$\frac{dx_{B2}}{dt} = \frac{F_1}{V_2}(x_{B1} - x_{B2}) + \frac{F_{20}}{V_2}(x_{B20} - x_{B2}) + \phi_c k_1 e^{\frac{-E_1}{rT_2}} x_{A2} - \phi_c k_2 e^{\frac{-E_2}{rT_2}} x_{B2} \qquad (20)$$

$$\frac{dT_2}{dt} = \frac{F_1}{V_2}(T_1 - T_2) + \frac{F_{20}}{V_2}(T_{20} - T_2) - \phi_c \frac{\Delta H_1}{c_p} k_1 e^{\frac{-E_1}{rT_2}} x_{A2} - \phi_c \frac{\Delta H_2}{c_p} k_2 e^{\frac{-E_2}{rT_2}} x_{B2} + \frac{Q_2}{\rho c_p V_2}$$

$$\frac{dx_{A3}}{dt} = \frac{F_2}{V_3}(x_{A2} - x_{A3}) - \frac{(F_r + F_p)}{V_3}(x_{Ar} - x_{A3})$$

$$\frac{dx_{B3}}{dt} = \frac{F_2}{V_3}(x_{B2} - x_{B3}) - \frac{(F_r + F_p)}{V_3}(x_{Br} - x_{B3})$$

$$\frac{dT_3}{dt} = \frac{F_2}{V_3}(T_2 - T_3) + \frac{Q_3}{\rho c_p V_3} + \frac{(F_r + F_p)}{\rho c_p V_3}(x_{Ar}\Delta H_{\text{vap1}} + x_{Br}\Delta H_{\text{vap2}} + x_{Cr}\Delta H_{\text{vap3}})$$

For a more detailed explanation of this process, refer to (Liu et al., 2008; Yin & Liu, 2019). Additional process disturbances are introduced into the systems to evaluate the robustness of the proposed control strategy. These disturbances are sampled from a multivariate normal distribution $\mathcal{N}(\mathbf{0}, \sigma_\epsilon^2)$, where $\sigma_\epsilon = [0.01, 0.01, 0.50, 0.01, 0.01, 0.50, 0.01, 0.01, 0.50]$ corresponds to the standard deviations associated with the nine state variables. For time-invariant RSCP, parameter $\phi_c$ is set as $\phi_c = 1$; For time-varying, parameter $\phi_c$ is set as $\phi_c = e^{-0.01t}$. The control objectives for both processes are to regulate the system states and ensure the states remain at a steady-state set point:

$$x_s = [0.18, 0.67, 480.32\,\text{K}, 0.20, 0.65, 472.79\,\text{K}, 0.07, 0.67, 474.89\,\text{K}]^{\text{T}}$$

Given a set of scaling coefficients $x_{scale} = [0.36, 0.18, 361.94\,\text{K}, 0.21, 0.18, 342.88\,\text{K}, 0.26, 0.21, 361.59\,\text{K}]^{\text{T}}$, the cost function for the two RSCP systems can be defined as:

$$cost_{RSCP}(x) = \left(\frac{x - x_s}{x_{scale}}\right)^2 \qquad (21)$$

## D   HYPERPARAMETERS

The hyperparameters for the MamKO, DKO, and MLP are listed in Table 4, Table 5, Table 6, respectively.

Table 4: Hyperparameters of MamKO

| Hyperparameters | Value |
|---|---|
| Size of training set | 40000 |
| Size of test set | 4000 |
| Batch Size | 256 |
| Learning rate | 1e-3 |
| Prediction horizon $H$ | 30 |
| Structure of $\psi$ (CartPole & Time-varying CartPole) | (4, 64, 8) |
| Structure of $\psi$ (GRN) | (6, 64, 10) |
| Structure of $\psi$ (Time-invariant RSCP & Time-varying RSCP) | (9, 64, 15) |
| Activation function in $\psi$ | ReLU |
| Kernel size $h$ of $\phi$ (CartPole) | 10 |
| Kernel size $h$ of $\phi$ (Time-varying CartPole) | 15 |
| Kernel size $h$ of $\phi$ (GRN) | 10 |
| Kernel size $h$ of $\phi$ (Time-invariant RSCP) | 10 |
| Kernel size $h$ of $\phi$ (Time-varying RSCP) | 5 |
| Dimension of observables (CartPole & Time-varying CartPole) | 8 |
| Dimension of observables (GRN) | 10 |
| Dimension of observables (time-invariant RSCP & Time-varying RSCP) | 15 |

Table 5: Hyperparameters of DKO

| Hyperparameters | Value |
|---|---|
| Size of training set | 40000 |
| Size of test set | 4000 |
| Batch Size | 256 |
| Learning rate | 1e-3 |
| Prediction horizon $H$ | 30 |
| Structure of $\psi$ (CartPole & Time-varying CartPole) | (4, 64, 8) |
| Structure of $\psi$ (GRN) | (6, 64, 10) |
| Structure of $\psi$ (Time-invariant RSCP & Time-varying RSCP) | (9, 64, 15) |
| Activation function | ReLU |
| Dimension of observables (CartPole & Time-varying CartPole) | 8 |
| Dimension of observables (GRN) | 10 |
| Dimension of observables (Time-invariant RSCP & Time-varying RSCP) | 15 |

Table 6: Hyperparameters of MLP

| Hyperparameters | Value |
|---|---|
| Size of training set | 40000 |
| Size of test set | 4000 |
| Batch Size | 256 |
| Learning rate | 1e-3 |
| Prediction horizon $H$ | 30 |
| Structure of $\phi$ (CartPole & Time-varying CartPole) | (4, 24, 16, 8) |
| Structure of $\phi$ (GRN) | (6, 30, 20, 10) |
| Structure of $\phi$ (Time-invariant RSCP & Time-varying RSCP) | (9, 45, 30, 15) |
| Activation function | ReLU |

# E    MODELING RESULTS

## E.1    COMPARISION WITH CLASSIC SYSTEM IDENTIFICATION METHOD

A classical system identification method – subspace identification (Van Overschee & De Moor, 2012) is also applied to build a data-based dynamic model for the dynamics of the CartPole system. For the 30-step-ahead prediction task, the model built based on subspace identification can only provide satisfactory predictions of the position and velocity of the cart, while the predictions of the state variables related to the pole diverge. In contrast, the proposed method can accurately forecast the future behaviors of all the states of the CartPole system.

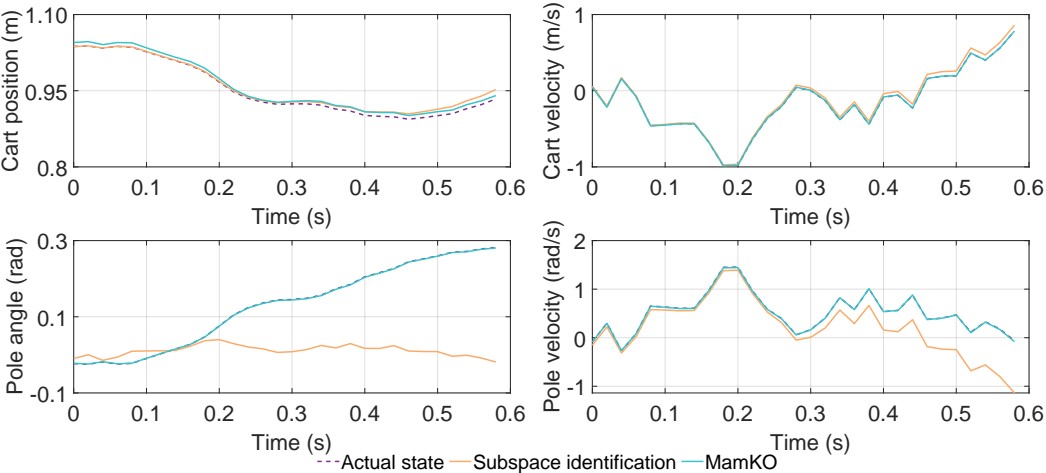

Figure 7: Modeling results of the CartPole system by MamKO and subspace identification.

## E.2    THE SELECTION OF THE DIMENSION OF THE STATE-SPACE MODEL

The dimension of the linear state-space model is determined through trial and error. A case study on the CartPole system is conducted to illustrate the selection of the dimension of the state-space model. The results, which include the test loss across various lifting dimensions, are presented in Table 7. As the dimension of the lifted space increases from 5 to 8, the test loss decreases significantly to $6.93 \times 10^{-4}$. Further increasing the dimension does not lead to significant improvement in the modeling performance.

Table 7: Test loss of different lifting dimensions on CartPole system modeling by MamKO.

| Lifting dimension | 5 | 8 | 10 | 15 |
|---|---|---|---|---|
| Test loss | $8.95 \times 10^{-4}$ | $6.93 \times 10^{-4}$ | $6.94 \times 10^{-4}$ | $7.04 \times 10^{-4}$ |

### E.3 PREDICTION ERROR ON EACH PREDICTION STEP

The modeling performance of the proposed method at each prediction step is evaluated. The results for the CartPole system, GRN system, and time-invariant RSCP system are presented in Figure 8, Figure 9, and Figure 10. From the results, it can be demonstrated that while the prediction error tends to increase as the number of prediction steps progresses, there is a notable reduction in error observed mid-way through the prediction horizon. This phenomenon underscores the effectiveness of our method in managing error propagation.

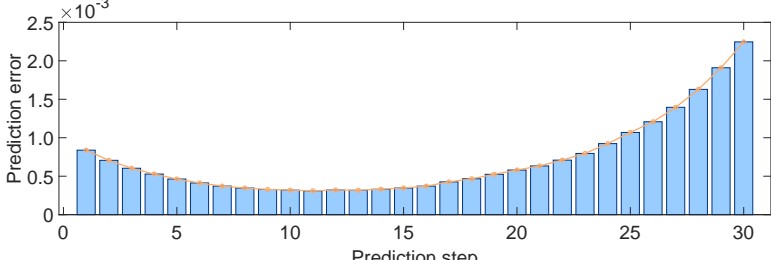

Figure 8: Prediction error of the CartPole system at each prediction step.

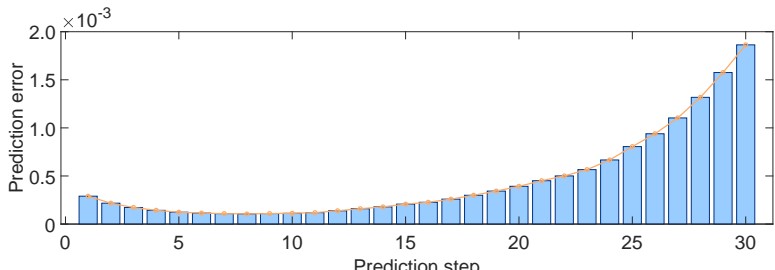

Figure 9: Prediction error of the GRN system at each prediction step.

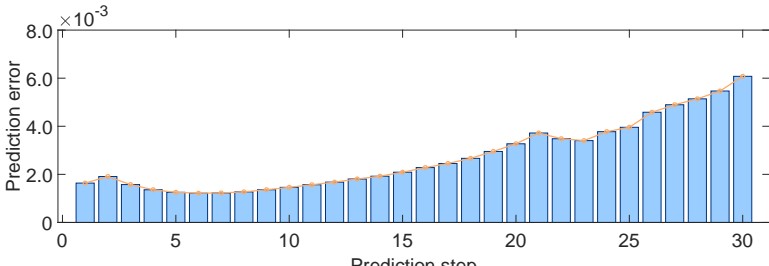

Figure 10: Prediction error of the time-invariant RSCP system at each prediction step.

## F  CONTROL EXPERIMENT RESULTS

### F.1  CARTPOLE AND TIME-VARYING CARTPOLE SYSTEMS

The trajectories of the $x$ and $\theta$ for the Cartpole systems are presented in Figure 11.

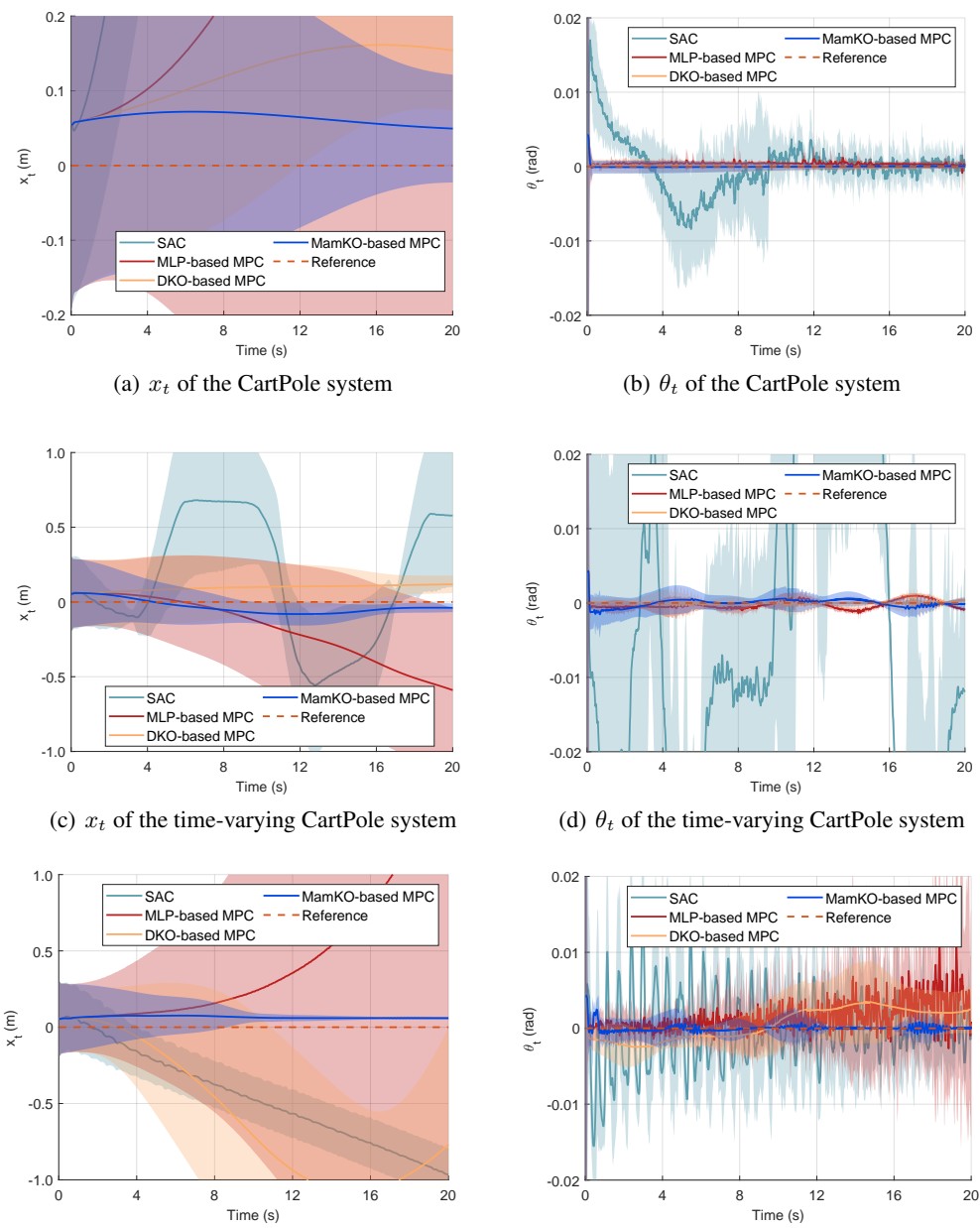

(a) $x_t$ of the CartPole system

(b) $\theta_t$ of the CartPole system

(c) $x_t$ of the time-varying CartPole system

(d) $\theta_t$ of the time-varying CartPole system

(e) $x_t$ of the time-varying CartPole system with rapidly changing parameters

(f) $\theta_t$ of the time-varying CartPole system with rapidly changing parameters

Figure 11: The trajectories of the Cartpole systems. The Y-axis indicates the average state trajectory of each time instant. The shaded area represents the confidence interval (one standard deviation) over the ten experiments.

The steady-state errors of the three CartPole systems are listed in Table 8. From the results, the MamKO-based MPC demonstrates the smallest steady-state errors in all the states of each of the three CartPole systems. Particularly, for the time-varying CartPole system with rapidly changing parameters, the MamKO-based MPC achieves a substantial reduction in the steady-state error in

cart position, outperforming the MLP-based MPC, DKO-based MPC and SAC by 98.38%, 92.92%, and 93.86%, respectively.

Table 8: Steady-state errors of the three CartPole systems by different control methods.

| Method | CartPole system | | Time-varying CartPole system | | CartPole system CartPole system with rapidly changing paramters | |
|---|---|---|---|---|---|---|
| | Cart position (m) | Pole angle (rad) | Cart position (m) | Pole angle (rad) | Cart position (m) | Pole angle (rad) |
| MamKO-based MPC | $6.40 \times 10^{-2}$ | $2.15 \times 10^{-4}$ | $3.98 \times 10^{-2}$ | $4.74 \times 10^{-3}$ | $5.94 \times 10^{-2}$ | $2.96 \times 10^{-4}$ |
| MLP-based MPC | 1.28 | $5.26 \times 10^{-4}$ | $6.62 \times 10^{-1}$ | $1.02 \times 10^{-3}$ | 3.66 | $1.03 \times 10^{-2}$ |
| DKO-based MPC | $1.54 \times 10^{-1}$ | $2.35 \times 10^{-4}$ | $1.30 \times 10^{-1}$ | $5.14 \times 10^{-3}$ | $8.39 \times 10^{-1}$ | $2.68 \times 10^{-3}$ |
| SAC | $6.94 \times 10^{-1}$ | $1.92 \times 10^{-2}$ | $7.34 \times 10^{-1}$ | $1.87 \times 10^{-2}$ | $9.67 \times 10^{-1}$ | $2.65 \times 10^{-2}$ |

## F.2 GRN SYSTEM

The trajectories of $x_4$ of the GRN system obtained based on different control methods are presented in Figure 12. The steady-state errors of the GRN system are listed in Table 9 .The proposed MamKO-based MPC provides the smallest steady-state error, reducing it by 52.23%, 92.10% and 90.68% as compared to MLP-based MPC, DKO-based MPC, and SAC, respectively.

Table 9: Steady-state error of the GRN system by different control methods.

| Method | MamKO-based MPC | MLP-based MPC | DKO-based MPC | SAC |
|---|---|---|---|---|
| Steady-state error | $5.35 \times 10^{-2}$ | $1.12 \times 10^{-1}$ | $6.77 \times 10^{-1}$ | $5.74 \times 10^{-1}$ |

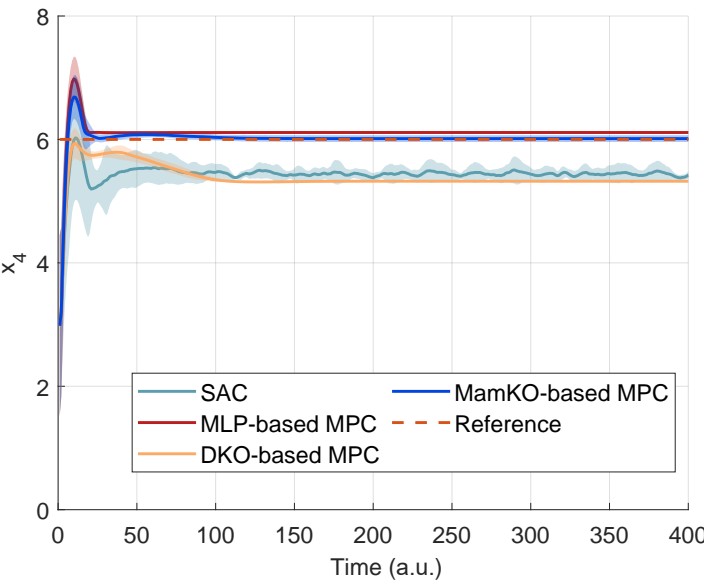

Figure 12: The trajectories of the GRN system by different control methods. The Y-axis indicates the average state trajectory of each time instant. The shaded area represents the confidence interval (one standard deviation) over the ten experiments

## F.3 TIME-INVARIANT RSCP AND TIME-VARYING RSCP

One set of trajectories obtained from ten experiments for the time-invariant RSCP system is presented in Figure 13. One set of trajectories obtained from the ten experiments for the time-varying RSCP system is presented in Figure 14. Since the nine states of the RSCP system have significantly different magnitudes, the tracking errors for the RSCP system are calculated based on the states after normalization. The steady-state errors of the time-invariant RSCP system and time-varying RSCP system are listed in Table 10. From the results, the proposed MamKO-based MPC demonstrates the smallest steady-state error. For the time-invariant RSCP system, the proposed MamKO-based MPC achieves reductions in the steady-state error by 57.33%, 17.42%, and 61.68% as compared to MLP-based MPC, DKO-based MPC, and SAC, respectively. For the time-varying RSCP system, the proposed MamKO-based MPC reduces steady-state error by 49.78%, 24.83%, and 22.25% as compared to MLP-based MPC, DKO-based MPC, and SAC, respectively.

Table 10: Steady-state errors of the time-invariant RSCP system and the time-varying RSCP system.

| Method | MamKO-based MPC | MLP-based MPC | DKO-based MPC | SAC |
|---|---|---|---|---|
| Time-invariant RSCP system | $1.28 \times 10^{-2}$ | $3.00 \times 10^{-2}$ | $1.55 \times 10^{-2}$ | $3.34 \times 10^{-2}$ |
| Time-varying RSCP system | $6.78 \times 10^{-2}$ | $1.35 \times 10^{-1}$ | $9.02 \times 10^{-2}$ | $8.72 \times 10^{-2}$ |

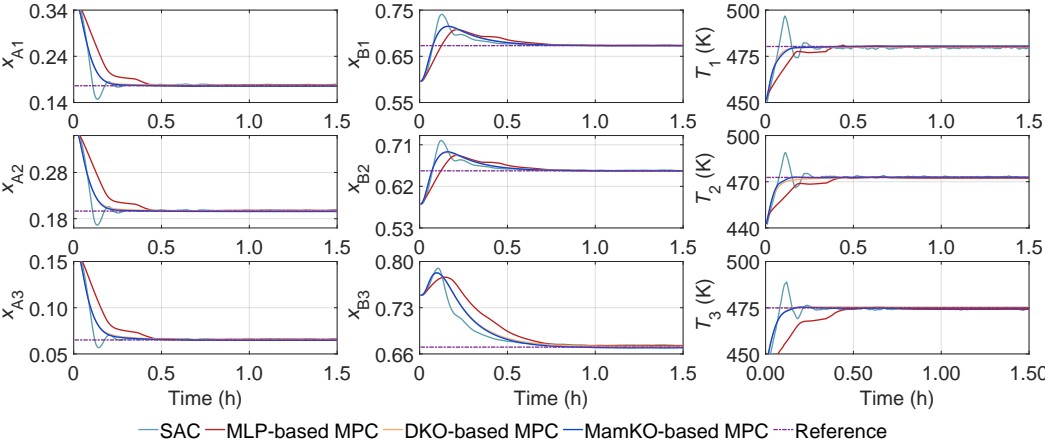

Figure 13: The state trajectories for the time-invariant RSCP systems.

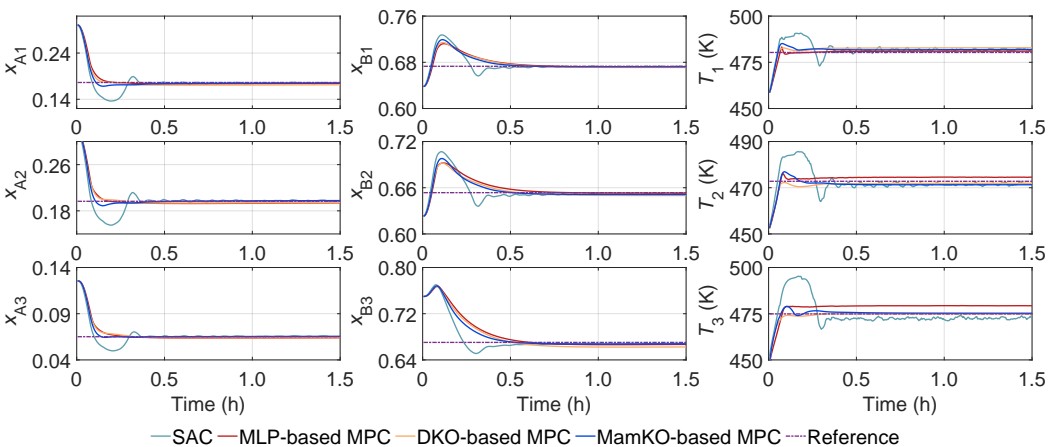

Figure 14: The state trajectories for the time-varying RSCP systems.

### F.4 WATEWATER TREATMENT SYSTEM

To further demonstrate the applicability of the proposed method on highly nonlinear systems, we evaluate the performance of MamKO on a large-scale nonlinear water treatment system (Alex et al., 2008), which contains 145 states, two control inputs and 14 known disturbances. The dynamic behaviors of this process are simulated using a high-fidelity simulator that was built based on 145 ordinary differential equations (ODEs) (Zeng & Liu, 2015). A schematic of the water treatment plant is presented in Figure 15.

15 invariant states are excluded from Koopman-based modeling and control tasks. The modeling task aims to build a Koopman model that describes the evolution of the remaining 130 state variables over a 10-step prediction horizon. For the water treatment system, trajectories of state and action samples are gathered, generating a training set of $20,000$ samples, a validation set of $2,000$ samples, and a test set of $2,000$ samples. The modeling performance of the MamKO method and two baseline methods are presented in Figure 16. From the experimental results, the proposed method outperforms the deep Koopman operator (DKO) and multilayer perception (MLP) methods in modeling performance.

A set-point tracking task is considered to evaluate the control performance of the proposed method. The objective of the control task is to drive two selected states to a desired set point under the influence of the disturbances. In the control task, a PID controller and an MPC based on the first-principles model denoted as NMPC are included as the baselines. From the experimental results, the proposed MamKO-based MPC reduces tracking error by $55.60\%$, $45.95\%$, and $10.07\%$ as compared to MLP-based MPC, DKO-based MPC, and SAC, respectively. The MPC based on the exact first-principles model (the same model as the simulator) exhibits the smallest tracking error. However, as emphasized in the introduction, obtaining an exact first-principles model can be challenging. Additionally, the nonlinear optimization required for first-principles model systems often results in suboptimal solutions, leading to large deviation points in the process. Our method achieves a reduction in tracking error of $11.56\%$ as compared to the PID controller. Furthermore, integrating state constraints with the PID controller for this system can be challenging, and switching to a different set point may require retuning the parameters.

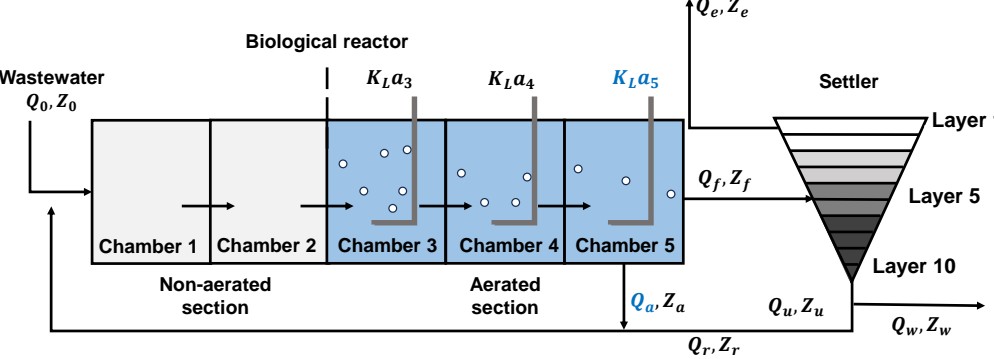

Figure 15: A schematic of the water treatment plant (Alex et al., 2008).

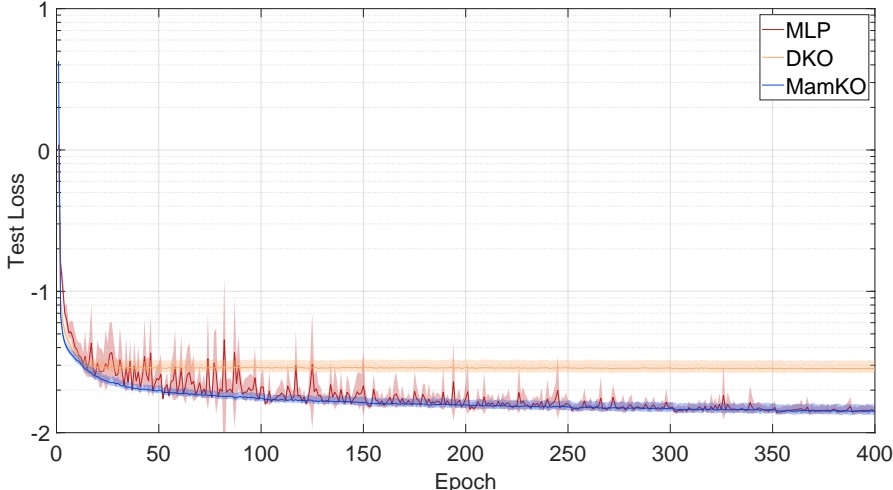

Figure 16: Test loss of water treatment process in log space from different methods. The shaded area represents the confidence interval (0.5 times the standard deviation) across ten training trials.

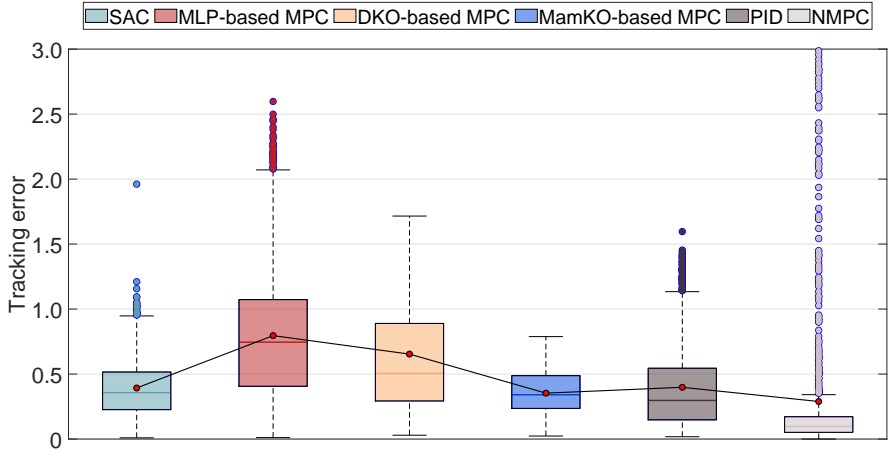

Figure 17: Tracking errors of the control task in the water treatment process from different methods. The red dots inside the boxes indicate the mean tracking errors in the process. The box chart shows the distribution of the tracking error. The middle line inside the box represents the median value of the collected data. The box represents the interquartile range (IQR), which encompasses the range between the first quartile (Q1) and the third quartile (Q3). The whiskers extend to the smallest and largest values within 1.5 times the IQR from the lower and upper quartiles. The outliers are the individual data points outside the whiskers.

## F.5 TIME-DELAY RSCP SYSTEM

From the application perspective of the RSCP system, the transportation lag of reactant $A$ and desired product $B$ may introduce time delays in measuring the mass fractions $x_{\text{Ai}}$ and $x_{\text{Bi}}$. In this system, a 0.025 h time delay is included. At each new sampling instant $k$, the controller optimizes control output based on the measurements of $x_{\text{Ai}}$ and $x_{\text{Bi}}$ at sampling instant $k - 5$, instantaneous measurements of $T_i$ and the historical trajectory. The modeling and control tasks of the time-delay RSCP system are the same as the time-invariant RSCP and time-varying RSCP system in Section 5. The modeling performance of each method is presented in Figure 18, and the control performance of each method is presented in Figure 19. From the results of modeling tasks, the proposed method reduces the test loss by 55.56% and 66.67% as compared with the MLP and DKO methods, respectively. For the control task, the proposed MamKO-based MPC reduces the cost by 52.52%, 53.59%, and 84.25% as compared to MLP-based MPC, DKO-based MPC, and SAC, respectively.

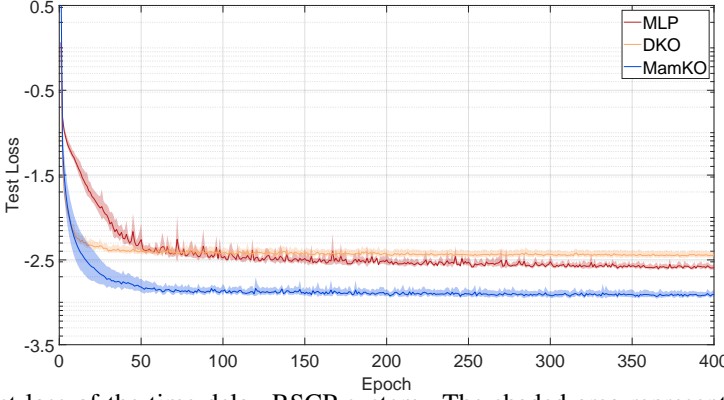

Figure 18: Test loss of the time-delay RSCP system. The shaded area represents the confidence interval (0.5 times the standard deviation) across ten training trials.

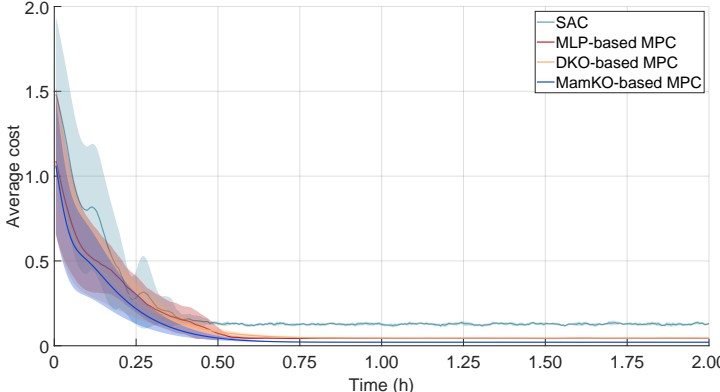

Figure 19: Average cost of the time-delay RSCP system in log space. The shaded area represents the confidence interval (0.3 times the standard deviation) over the ten experiments.

### F.6 TRAJECTORY TRACKING TASK FOR GRN SYSTEM

A sinusoidal function with periods of 400 seconds is set as the tracking target of the task, which can be represented as:

$$x(t) = 5 \sin\left(\frac{\pi}{200}t\right) + 10 \tag{22}$$

In the tracking process, extra uniform disturbance ranging from $[-0.5, 0.5]$, i.e., $\xi_i \sim \mathcal{U}(-0.5, 0.5)$ is added in the tracking process. The tracking trajectories of the GRN system are shown in Figure 20. The mean tracking errors of the different methods are presented in Table 11. From the results, the MamKO-based MPC exhibits the best tracking performance among the control methods, which reduces the tracking error by $9.84\%$, $20.29\%$, $32.93\%$ as compared to MLP-based MPC, DKO-based MPC, and SAC, respectively. The experimental results demonstrate that the proposed method can be applied to trajectory-tracking tasks.

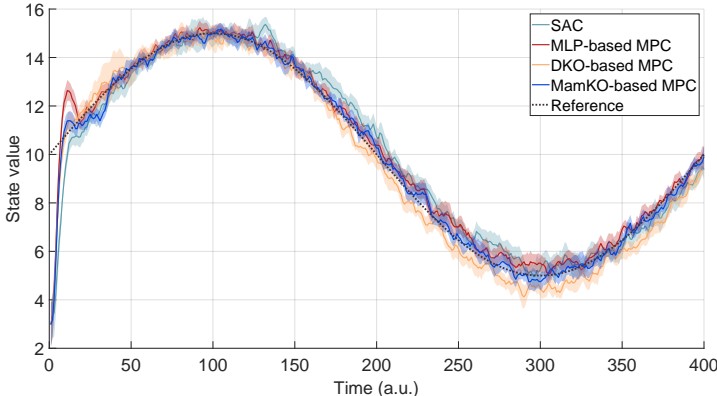

Figure 20: Tracking trajectories of the GRN system of different methods. The Y-axis indicates the average state trajectory of each time instant. The shaded area represents the confidence interval (one standard deviation) over the ten experiments.

Table 11: Mean tracking errors of GRN system of different methods.

| System | MamKO-based MPC | MLP-based MPC | DKO-based MPC | SAC |
|---|---|---|---|---|
| Mean tracking error | 0.55 | 0.61 | 0.69 | 0.82 |

