# OpenReview forum: "MamKO: Mamba-based Koopman operator for modeling and predictive control"
_ICLR.cc/2025/Conference — ICLR 2025 Spotlight_

### Official Review · Reviewer_PG54 · 2024-11-02

**Soundness:** 3
**Presentation:** 3
**Contribution:** 4
**Rating:** 8
**Confidence:** 4

**Summary:**

The paper innovatively combines the Mamba architecture with Koopman operators, introducing the MamKO framework - the first application of a large model architecture to Koopman modeling and control. The authors effectively leverage Mamba's capabilities in time-varying modeling and improve the matrix generation mechanism by replacing the negative exponential function with CELU activation function, enabling the model to handle unstable systems. Furthermore, based on the generated time-varying Koopman operator, they designed an MPC controller that significantly enhances modeling and control performance while maintaining computational efficiency. In the experimental section, the authors demonstrate MamKO's practical performance across multiple application scenarios (such as CartPole and chemical processes) and compare it with other control methods (like SAC) to validate its superiority.

**Strengths:**

- Innovation: First-time integration of Mamba architecture with Koopman operators, introducing the MamKO framework, pioneering a new application of large models in control theory.

- Performance: Demonstrated superior performance of MamKO-MPC across multiple practical application scenarios, with effective comparisons against existing control methods.

- Model Enhancement: Improved the matrix generation mechanism in Mamba through CELU activation function, enabling the handling of unstable systems and enhancing model robustness.

- Computational Efficiency: The designed MPC controller maintains high computational efficiency while improving modeling and control performance, reducing the computational burden associated with traditional online Koopman methods.

**Weaknesses:**

The paper has a clear overall structure, but lacks sufficient depth in key areas, particularly in justifying the choice of Mamba over other sequence model architectures. Furthermore, the limited explanation of theoretical background makes it challenging for readers to fully understand the design rationale and advantages of the MamKO framework. Specifically:

Theoretical Analysis Needs Strengthening:
- Lacks in-depth theoretical analysis of system stability and convergence. As a control method incorporating deep learning, it should provide more comprehensive theoretical guarantees for system stability

Insufficient Model Selection Justification:
- While the similarity between Mamba's SSM structure and Koopman operators is mentioned, a detailed mathematical comparison between the Mamba SSM structure and Koopman operators lacks and reasons for choosing Mamba over other sequence model architectures based on SSM (like S4, H3) inadequate

Limited Control Task Types:
- Control tasks primarily focus on setpoint tracking, missing evaluation of other control tasks (trajectory tracking, disturbance rejection, energy optimization)

Experimental Evaluation Metrics Need Expansion:
- Lacks systematic evaluation of control performance metrics (overshoot, steady-state error), recommend including more control performance indicators to comprehensively demonstrate MamKO-MPC's advantages

**Questions:**

To enhance the impact and persuasiveness of this work, the following questions merit consideration:

1. A detailed analysis or proof regarding the theoretical stability and convergence properties of the MamKO framework would be valuable.

2. Validation of selecting the Mamba structure over other SSM-based models (such as S4, H3) would strengthen the paper, along with elaboration on Mamba's mathematical advantages and rationale in control tasks.

3. Could more evidence about the MamKO framework's generalization ability and robustness when handling systems with different dynamic characteristics be supported by experimental data?

---

> ### Author Response · Authors · 2024-11-28
> **Responses to Reviewer PG54**
>
> We greatly appreciate the reviewer’s insightful questions and suggestions. We have revised the manuscript according to the reviewer’s suggestions.
>
> 1. Weakness 1: Lacks in-depth theoretical analysis of system stability and convergence. As a control method incorporating deep learning, it should provide more comprehensive theoretical guarantees for system stability. Question 1: A detailed analysis or proof regarding the theoretical stability and convergence properties of the MamKO framework would be valuable.
>
> We thank the reviewer for the insightful comment.
> The primary objective is to leverage the integration of the Mamba and Koopman operators to develop a unified modeling and control approach for general nonlinear systems with time-varying parameters. We agree with the reviewer that a formal theoretical analysis of the convergence and stability of the proposed approach will be instrumental in ensuring the proposed method can achieve the desired prediction and control performance. Meanwhile, the stability analysis for the proposed control method, which is built based on time-varying Koopman operators, is challenging due to several factors. First, the underlying nonlinear system features time-varying parameters. Second, the use of time-varying Koopman operators in modeling further complicates the theoretical analysis. Third, there exists a plant-model mismatch between the real-world plant and the established MamKO model. Therefore, formal theoretical analysis of convergence and stability properties is left to our future work.
>
> In response to this comment, as well as Comment #7 from Reviewer A4oR and Comment #3 from Reviewer 41k8, we have added a sentence at the end of Section 6 to discuss this future research direction.
>
> 2. Weakness 2: While the similarity between Mamba's SSM structure and Koopman operators is mentioned, a detailed mathematical comparison between the Mamba SSM structure and Koopman operators lacks and reasons for choosing Mamba over other sequence model architectures based on SSM (like S4, H3) inadequate.
> Question  2: Validation of selecting the Mamba structure over other SSM-based models (such as S4, H3) would strengthen the paper, along with elaboration on Mamba's mathematical advantages and rationale in control tasks.
>
> We thank the reviewer for the comment. As noted in the introduction, a key factor in selecting Mamba is its capability to effectively represent time-varying systems, facilitating the formulation of a time-varying state-space model as outlined in Eq. (5) of the revised manuscript. While the S4 and H3 models have demonstrated impressive performance in language modeling, the state-space models (SSMs) utilized within both remain time-invariant. The linear state-space models in these methods are similar to the Koopman model described in Eq. (3) of the revised manuscript. We evaluate the modeling and control performance of the DKO method, which utilizes a time-invariant linear state-space model, and found that it provides less satisfactory results compared to the MamKO method. Overall, for time-varying systems, models derived from time-varying state-space models tend to produce better results, and we choose the Mamba structure accordingly.
>
> In response to this comment, we have added a sentence in the third paragraph of the related work to discuss the features of Mamba.

---

> ### Author Response · Authors · 2024-11-28
> **Responses to Reviewer PG54**
>
> 3. Weakness 3:  Control tasks primarily focus on setpoint tracking, missing evaluation of other control tasks (trajectory tracking, disturbance rejection, energy optimization).
>
> We thank the reviewer for the insightful comment. We consider the set-point tracking task since it is one of the most common and fundamental control tasks.
> The set-point tracking tasks of the reactor-separator chemical process (RSCP) system and its variant in this paper are under the intervention of process disturbances. The process disturbances are generated following a multivariate normal distribution $\mathcal{N}(\mathbf{0},\sigma_\epsilon^2)$ with $\sigma_\epsilon=\left[0.01, 0.01, 0.50, 0.01, 0.01, 0.50, 0.01, 0.01, 0.50\right]$ for the nine state variables. From the experimental results in  Figure 2 and Figure 3 of the revised manuscript, the proposed method demonstrates superior performance in both modeling and control, demonstrating the ability to reject disturbances.
>
> For the trajectory tracking task, we add the task for the gene regulatory networks (GRN) system in F.6 of the Appendix. A sinusoidal function with periods of 400 seconds is set as the tracking target of the task.
> In the tracking process, extra uniform disturbance ranging from $[-0.5,0.5]$, i.e., $\xi_i\sim \mathcal{U}(-0.5,0.5)$ is added to further check the robustness of the system.
> From the results in Figure 20 and Table 11 of the Appendix, the MamKO-based model predictive controller (MPC) exhibits the best tracking performance among the baseline methods, which reduces the tracking error by 9.84%, 20.29%, 32.93% as compared to MLP-based MPC, DKO-based MPC, and SAC, respectively. The results demonstrate that the proposed method can be applied to trajectory-tracking tasks.
>
> The energy optimization tasks are also important control tasks, and the proposed method has the potential to be extended to handle these control tasks. For example, we are considering optimal control of a membrane-based reactor to reduce its energy consumption. The energy optimization task will be achieved by developing an economic MPC scheme [1-2]  based on MamKO modeling. We will investigate this problem systematically in future research.
>
> In response to this comment, We have added a trajectory tracking task for the GRN system in F.6 of the Appendix.
> We have added a sentence at the end of Section 6 to discuss this future research direction.
>
> 4. Weakness 4: Lacks systematic evaluation of control performance metrics (overshoot, steady-state error), recommend including more control performance indicators to comprehensively demonstrate MamKO-MPC's advantages.
>
> We thank the reviewer for the suggestion about the evaluation metrics of the control performance. Due to space constraints, we relocate part of the control results to Appendix F, where readers can view the trajectories of the control tasks.
>
> Following this comment, we compute the steady-state errors of the six benchmark systems, and the results are presented in Table 8, Table 9, and Table 10.
> The steady-state errors of the three CartPole systems are listed in Table 8 of the Appendix. From the results, the MamKO-based MPC demonstrates the smallest steady-state errors in all the states of each of the three CartPole systems. Particularly, for the time-varying CartPole system with rapidly changing parameters, the MamKO-based MPC achieves a substantial reduction in the steady-state error in cart position, outperforming the MLP-based MPC, DKO-based MPC and SAC by 98.38%, 92.92%, and 93.86%, respectively. The steady-state errors of the GRN system are listed in Table 9 of the Appendix. The proposed MamKO-based MPC provides the smallest steady-state error, reducing it by 52.23%, 92.10%, and 90.68% as compared to MLP-based MPC, DKO-based MPC, and SAC, respectively. The steady-state errors of the RSCP system and time-varying RSCP system are listed in Table 10 of the Appendix. From the results, the proposed MamKO-based MPC demonstrates the smallest steady-state error. For the RSCP system, the proposed MamKO-based MPC achieves reductions in the steady-state error by 57.33%, 17.42%, and 61.68% as compared to MLP-based MPC, DKO-based MPC, and SAC, respectively. For the time-varying RSCP system, the proposed MamKO-based MPC reduces steady-state error by 49.78%, 24.83%, and 22.25% as compared to MLP-based MPC, DKO-based MPC, and SAC, respectively.
>
> In response to this comment, we have added Table 8, Table 9, and Table 10 in the Appendix to present the control performance metrics.
>
> [1] J. B, Rawlings, D. Angeli, C. Bates, Fundamentals of Economic Model Predictive Control. IEEE Conference on Decision and Control, 3851-3861, Maui, HI, USA, 2012.
>
> [2] M. Ellis, H. Durand, P.D. Christofides, Fundamentals of Economic Model Predictive Control. Journal
> of Process Control , 24(8): 1156-1178, 2014.

---

> > ### Author Response · Authors · 2024-11-28
> > **Responses to Reviewer PG54**
> >
> > 5. Question 3: Could more evidence about the MamKO framework's generalization ability and robustness when handling systems with different dynamic characteristics be supported by experimental data?
> >
> > We thank the reviewer for the question about the generalization ability and robustness. For the generalization ability of the proposed framework, we consider the different benchmark systems across various industries to demonstrate the generalization ability of the proposed method. For example, the CartPole system with high-frequency dynamics and the RSCP system with low-frequency dynamics are all considered in this paper. From the results presented in Figure 2 and Figure 3 of the revised manuscript, the MamKO method can achieve stable modeling and control performance across all the benchmark systems. Especially for the control performance, the MamKO-based MPC demonstrates superior control performance compared to the baselines.
> >
> > For the robustness of the system,  the reactor-separator chemical process (RSCP) system and its variant in this paper are in the presence of disturbances. The process disturbances are generated following a multivariate normal distribution $\mathcal{N}(\mathbf{0},\sigma_\epsilon^2)$ with $\sigma_\epsilon=\left[0.01, 0.01, 0.50, 0.01, 0.01, 0.50, 0.01, 0.01, 0.50\right]$ for the nine state variables. From the experimental results in  Figure 2 and Figure 3 of the revised manuscript, the proposed method demonstrates superior performance in both modeling and control, demonstrating the robustness of the proposed method.
> >
> > We thank the reviewer again for all the comments.

---

### Official Review · Reviewer_41k8 · 2024-11-02

**Soundness:** 3
**Presentation:** 3
**Contribution:** 3
**Rating:** 6
**Confidence:** 4

**Summary:**

The paper investigates complex dynamical systems and aims to address limitations in modeling these systems using Koopman operators. It explores the application of large language model (LLM) technology to manage the nonlinear and time-varying characteristics of dynamic systems. The work introduces a framework combining Mamba with Koopman operators, leveraging Mamba’s matrix generation capabilities to predict the future states of time-varying systems. This approach preserves a linear state-space structure while enhancing the model’s predictive accuracy and adaptability, creating a linear invariant state-space model. The paper is well-structured and logically organized, presenting innovative ideas with strong experimental support and highlighting the potential impact of this approach on complex dynamical system modeling and control.

**Strengths:**

1. The paper provides comprehensive experimental steps, data, and ablation analyses to validate the superiority of the MamKO model.
2. Through experiments on multiple benchmark systems, the paper demonstrates the advantages of MamKO in handling time-varying systems.
3. The paper shows that MamKO modeling relies on multi-step prediction, enabling the model to scale to complex systems with multi-step dependencies effectively.
4. Based on the MamKO model, the paper develops an MPC (Model Predictive Control) scheme, emphasizing its computationally efficient optimal control for nonlinear systems.

**Weaknesses:**

1. The paper does not provide quantitative metrics to justify why the Mamba model was chosen over other models, nor does it include comparative experiments with other LLM models.
2. All experiments on time-varying systems were conducted in simulated environments, lacking consideration of real-world physical factors. Thus, there are questions regarding the actual performance of the MamKO model.
3. Although the paper emphasizes solving time-varying problems in control systems, it lacks an analysis or discussion of the control scheme’s stability and robustness.
4. The proposed approach uses multi-step historical data but needs to address the accumulation of errors that this may cause, which significantly affects the rigor of the model.
5. While the paper aims to apply the MamKO framework to time-varying systems to address related issues, it does not explicitly discuss the delay errors present in time-varying systems or propose corresponding solutions.

**Questions:**

1. The paper chooses the Mamba model as the basis for the overall approach but needs to provide quantifiable evidence supporting this choice. Could the authors provide relevant data to clarify the rationale for choosing the Mamba model?
2. All validation experiments were conducted on a simulation platform, with no hardware experiments presented. Simulation results are not a substitute for results from real-world experiments. Could the authors conduct hardware experiments to verify the feasibility and effectiveness of the approach?
3. The proposed framework aims to solve time-varying issues in time-varying systems but needs more discussion of delay errors and accumulated multi-step prediction errors, as well as the effects these errors may have on system stability and robustness. Did the authors consider these issues in the initial design phase? If not, could they now provide evidence to show the approach can address these issues?
4. The paper indicates that MamKO relies on multi-step prediction to achieve scalability for complex systems. Could you provide a detailed comparison of the computational overhead between MamKO and other methods? For instance, do quantitative data on computation time or hardware resource consumption for equivalent tasks exist to demonstrate the efficiency of MamKO?

---

> ### Author Response · Authors · 2024-11-28
> **Responses to Reviewer 41k8**
>
> We greatly appreciate the reviewer’s insightful questions and suggestions. We have revised the manuscript according to the reviewer’s suggestions.
>
> 1. Weakness 1: The paper does not provide quantitative metrics to justify why the Mamba model was chosen over other models, nor does it include comparative experiments with other LLM models. Question 1: The paper chooses the Mamba model as the basis for the overall approach but needs to provide quantifiable evidence supporting this choice. Could the authors provide relevant data to clarify the rationale for choosing the Mamba model?
>
> We thank the reviewer for the question. As discussed in the introduction, this paper focuses on two key aspects: modeling performance and real-time control efficiency. For large language models (LLMs) based on Transformer, they can achieve high modeling accuracy with the help of a large number of parameters and carefully designed structures. However, these features also impede their applicability to model predictive controller designs, as the complexity of the structure may complicate the optimization process.
> To demonstrate the impracticability of applying other LLMs on control systems, we conduct simulations of the computation times of the baselines on the benchmark systems.
> In Table 2 of the revised manuscript, the computation time of each baseline method is included. In Table 3 of the revised manuscript, the details of the sampling periods of the benchmark systems are included to illustrate the real-time requirements of these systems.
> The model predictive controller (MPC) based on a three-layer multilayer perception (MLP) requires $7.43\times 10^{-1}$ s to generate the control signal for the CartPole system, while the sampling periods for the system is 0.02s. The results also indicate that the extended computation time required by the MLP-based MPC, which is due to the need to solve nonlinear optimization problems, poses challenges for its online implementation. This limitation is particularly critical for systems with fast dynamics and short sampling periods, and for larger-scale systems with numerous control inputs and state variables to optimize.
> For larger models that are composed of considerably more parameters, the real-time applicability of the designed controllers would be much more severe. Conversely, the Mamba structure facilitates the straightforward implementation of a linear state-space model within a convex optimization-based MPC framework, thereby guaranteeing the real-time applicability of the control scheme.
>
> In response to this comment, we have added explanations with the simulation results in the second paragraph of Section 5.5 of the revised manuscript to discuss the impracticability of applying other LLM models.
>
> 2.  Weakness 2: All experiments on time-varying systems were conducted in simulated environments, lacking consideration of real-world physical factors. Thus, there are questions regarding the actual performance of the MamKO model.  Question 2: All validation experiments were conducted on a simulation platform, with no hardware experiments presented. Simulation results are not a substitute for results from real-world experiments. Could the authors conduct hardware experiments to verify the feasibility and effectiveness of the approach?
>
> We thank the reviewer for the question. To demonstrate the applicability of the proposed method on real-world systems, we evaluate the performance of MamKO on a large-scale nonlinear water treatment system [1] in F.4 of the Appendix, which contains 145 states, two control inputs, and 14 known disturbances. The dynamic behaviors of this process are simulated using a high-fidelity simulator that was built based on 145 ordinary differential equations (ODEs) [2].
> From the experimental results in Figure 16 of the revised manuscript, the proposed method outperforms the DKO and MLP methods in modeling performance. Additionally, in a set-point tracking task, the MamKO-based MPC exhibited superior control performance compared to soft actor-critic (SAC), deep Koopman operator (DKO)-based MPC, and MLP-based MPC. From the experimental results in Figure 17 of the revised manuscript, the proposed MamKO-based MPC reduced tracking error by 55.60%, 45.95%, and 10.07% as compared to MLP-based MPC, DKO-based MPC, and SAC, respectively. We agree with the reviewer that applying the approach in a real-world system is appealing, and we are currently conducting real-world experiments on a water treatment system, of which the results will be included in our future work.
>
> [1] J. Alex, L. Benedetti, J. Copp, K. Gernaey, U. Jeppsson, I. Nopens, M. Pons, L. Rieger, C. Rosen, J. Steyer, et al. Benchmark simulation model No. 1 (BSM1). Report by the IWA Task group on benchmarking of control strategies for WWTPs, 1, 2008.
>
> [2] J. Zeng and J. Liu. Economic model predictive control of wastewater treatment processes. Industrial & Engineering Chemistry Research, 54(21):5710–5721, 2015.

---

> > ### Author Response · Authors · 2024-11-28
> > **Responses to Reviewer 41k8**
> >
> > 3. Weakness 3: Although the paper emphasizes solving time-varying problems in control systems, it lacks an analysis or discussion of the control scheme’s stability and robustness.
> >
> > We thank the reviewer for the insightful comment.
> > The primary objective is to leverage the integration of the Mamba and Koopman operators to develop a unified modeling and control approach for general nonlinear systems with time-varying parameters. We agree with the reviewer that a formal analysis of the stability of the proposed approach will be instrumental in ensuring the proposed method can achieve the desired prediction and control performance. Meanwhile, the stability analysis for the proposed control method, which is built based on time-varying Koopman operators, is challenging due to several factors. First, the underlying nonlinear system features time-varying parameters. Second, the use of time-varying Koopman operators in modeling further complicates the theoretical analysis. Third, there exists a plant-model mismatch between the real-world plant and the established MamKO model. Therefore, formal theoretical analysis of stability properties is left to our future work.
> >
> > Regarding the robustness of the proposed scheme, empirical evaluation is conducted in the paper.
> > For the reactor-separator chemical process (RSCP) and its time-varying variant, extra process disturbances are added to the systems to test the robustness of the proposed control method. The disturbances are generated following a multivariate normal distribution $\mathcal{N}(\mathbf{0},\sigma_\epsilon^2)$ with $\sigma_\epsilon=\left[0.01, 0.01, 0.50, 0.01, 0.01, 0.50, 0.01, 0.01, 0.50\right]$ for the nine state variables.
> > The results in Figure 3 of the manuscript demonstrate that our proposed method can mitigate the impact of disturbances and consistently achieve the tracking task with the lowest error, which underscores the effectiveness of our approach in maintaining robust control across various dynamic and challenging environments.
> >
> > In response to this comment, as well as Comment #7 from Reviewer A4oR and Comment #1 from Reviewer PG54, we have added a sentence at the end of Section 6 to discuss this future research direction.
> > In response to this comment, we have added the explanation in C.3 of the Appendix to illustrate the disturbances that are added to the RSCP system.
> >
> > 4. Weakness 4: The proposed approach uses multi-step historical data but needs to address the accumulation of errors that this may cause, which significantly affects the rigor of the model. Question 3: The proposed framework aims to solve time-varying issues in time-varying systems but needs more discussion of delay errors and accumulated multi-step prediction errors, as well as the effects these errors may have on system stability and robustness. Did the authors consider these issues in the initial design phase? If not, could they now provide evidence to show the approach can address these issues?
> >
> > We appreciate the insightful question. MamKO utilizes a convolutional layer to extract hidden information from historical data, facilitating the prediction of future states. Unlike sequential processing methods such as LSTM, which process data step-by-step, the convolutional layer simultaneously extracts information from each historical step. Consequently, this method minimizes the risk of error accumulation through the historical time steps, even if the historical data is corrupted by noise.
> >  During the training process, we formulated the optimization from the multi-step-ahead prediction task in Eq. (9) of the revised manuscript, and prediction errors are calculated by the prediction error of each step without preference. Compared to the methods that only focus on one-step-ahead prediction accuracy, the proposed method can better reduce the accumulative error. We evaluate the modeling performance of the proposed method at each prediction step for the CartPole system, GRN system, and RSCP system, with results presented in Figure 11, Figure 12, and Figure 13 of the Appendix, respectively. From the results, we observe that while the prediction error tends to increase as the number of prediction steps progresses, there is a notable reduction in error observed mid-way through the prediction horizon. This phenomenon underscores the effectiveness of our method in managing error propagation.
> >
> > In response to this comment, we have added Figure 11, Figure 12, and Figure 13 in E.3 of the Appendix to show the prediction error on each prediction step.

---

> > > ### Author Response · Authors · 2024-11-28
> > > **Responses to Reviewer 41k8**
> > >
> > > 5. Weakness 5:  While the paper aims to apply the MamKO framework to time-varying systems to address related issues, it does not explicitly discuss the delay errors present in time-varying systems or propose corresponding solutions.
> > >
> > > We thank the reviewer for the valuable comment. Different from the traditional system identification methods that require prior information about system delays, MamKO does not need such information when establishing the linear state-space model. MamKO extracts the system dynamics from historical data, potentially enabling it to accommodate and analyze time delays, especially when the historical sequence is sufficiently long.
> > > To demonstrate that the proposed method can effectively manage time-delay systems, we introduce a 5-step time delay in the RSCP system in F.5 of the Appendix and compare the performance of our method against other baseline methods. Figure 18 and Figure 19 of the Appendix present the modeling and control performance of the MamKO method in the time-delay RSCP system.  From the results of modeling tasks, the proposed method reduces the test loss by 55.56%, 66.67% as compared with the MLP and DKO methods, respectively. For the control task, the proposed MamKO-based MPC reduced the cost by 52.52%, 53.59%, and 84.25% as compared to MLP-based MPC, DKO-based MPC, and SAC, respectively.
> > > This comparison highlights our method's robustness and capability to adapt to systems with time delays.
> > >
> > > In response to this comment, we have added simulations on the time-delay RSCP system in F.5 of the Appendix to test the modeling and control performance of the proposed method.
> > >
> > > 6. Question 4: The paper indicates that MamKO relies on multi-step prediction to achieve scalability for complex systems. Could you provide a detailed comparison of the computational overhead between MamKO and other methods? For instance, do quantitative data on computation time or hardware resource consumption for equivalent tasks exist to demonstrate the efficiency of MamKO?
> > >
> > > We thank the reviewer for the valuable question.
> > > We evaluate the computation times of the control methods on different benchmark systems. The online control implementation of the control methods is conducted on a computer equipped with an Intel Core i9-13900K CPU and 128 GB DDR4 RAM. The computation times of the MamKO-based MPC and baselines during control on the benchmark systems are presented in Table 2 of the revised manuscript. Compared with the MLP-based MPC, the MamKO-based MPC reduces the computation time by 98.38%, 83.39%, 99.21%, 90.34%, and 99.17% for the five benchmark systems, respectively. The efficient online implementation of the proposed method stems from the use of a linear state-space model within the proposed framework, which facilitates the formulation of an optimal control problem that requires solving convex optimization despite the nonlinearity in the dynamics of the considered systems.
> > >
> > > Following the reviewer's comment, as well as Comment #3 from Reviewer 9kPH and Comment #6 from
> > > Reviewer A4oR, we have added Table 2 of the revised manuscript to present the computation times for each method applied to the benchmark systems.

---

### Official Review · Reviewer_A4oR · 2024-11-02

**Soundness:** 3
**Presentation:** 3
**Contribution:** 3
**Rating:** 8
**Confidence:** 5

**Summary:**

the paper MamKO leverages Mamba architecture to generate koopman operators  with projected benefits towards model predication capability and adaptability relative to koopman models with constant koopman operators.  MamKO generates the Koopman Operator from realtime data making the model learn nonlinear behavior over time. To demonstrate capability the MamKO model is applied to model predictive control (MPC).  The modeling and control performance of the proposed method is evaluated through experiments on benchmark time-invariant and time-varying systems.
Four claims are made as advancing the state of the art
1. Matrix generation using Mamba structure
2. approach can handle unstable and time varying systems
3. Computationally efficient MPC is developed based on the MamKO approach
4. Experiments are provided in support.
Overall, the paper is well written and the developments are clear and error free.  While the authors have not demonstrated the results against any real world practical system, they do provide adequate development for a practitioner to adopt into real world problems and study applicability.  In that regard I think this work is worth publishing so the SOTA in Kooman based approaches   can continue to be explored for real world use.

**Strengths:**

the paper has several strengths
1. Leveraging Mamba for generating time-varying Koopman operators to address the limitations of fixed linear state-space models is innovative.
2. The authors  effectively incorporate MamKO into an MPC framework and demonstrate practical applications.  ability to formulate the control problem as a convex optimization task adds to the framework's appeal for real-time applications.
3. The experimental validation (although all simulations)  is based on multiple benchmark systems (e.g., CartPole, Gene Regulatory Networks, and CSTRs). Results demonstrate the advantages of MamKO over traditional methods like Deep Koopman Operator (DKO) and MLP-based models.
4. Real-Time Adaptability: By generating matrices in real-time, MamKO addresses the challenge of modeling time-varying systems efficiently. The proposed method's performance on rapidly changing systems highlights its practical utility.
5. Supplemental material allows reproducibility

**Weaknesses:**

The framework assumes that the Mamba-generated matrices adequately capture the dynamics of the underlying system. This assumption may not hold for  highly nonlinear systems where matrix representations could become unstable or inaccurate.

Experimental results do support the claims of improved performance. However, the paper could enhance its rigor by including statistical analyses (e.g., confidence intervals or hypothesis tests) to demonstrate the significance of the results.

Authors do acknowledge that their method may not generalize to all real-world systems, but a more explicit discussion on the boundaries of applicability (e.g., systems with extremely high-frequency dynamics) would add value.

Comparison: The benchmark methods used have all been solved using classical/modern control methods, yet eth author did not include any non data-driven solutions for comparison.  This may be a useful addition to the paper.

**Questions:**

Recommended fixes:
which provides enhanced model predictability and adaptability, as compared to Koopman models with constant Koopman operators -- you mean prediction capability  ?
Since time scales are important , figures should include time , instead of time steps ( or epochs) , Fig 8 just has  "t" with no units,  Please provide unit son all figure axes where possible

Claims requiring more evidence:
MamKO achieves good balance between advanced modeling capabilities and real-time control implementation efficiency -->
This claim in RT implementation needs more support hence actual time units are needed.  The RSCP problem has a scale of hours and is therefore generally very slow.  It will be good to see clearly the range of problems in time scale and frequency scale to understanding capability against system dynamics.

Theoretical:  The paper could use more rigorous theoretical analysis or proofs to back the claims of stability and effectiveness in capturing time-varying dynamics. For example, formal proofs of the convergence properties of the Mamba-based Koopman operators or bounds on the error in approximating nonlinear systems would greatly increase the paper’s credibility.

Equation (5): The transition from a time-invariant to a time-varying state-space model is theoretically justified but lacks a rigorous mathematical proof. The paper relies heavily on empirical evidence, perhaps some theoretical discussion may be help support eth generalizability of the method.

CELU Activation: The use of the negative CELU function is an interesting choice to handle instability. The mathematical justification provided for this choice seems reasonable, but more in-depth analysis of how it impacts model stability across different systems would strengthen the claim.

**Details Of Ethics Concerns:**

no concerns on ethics

---

> ### Author Response · Authors · 2024-11-28
> **Responses to Reviewer A4oR**
>
> Thank you very much for the detailed comments, which are very helpful to us. We have revised the manuscript according to the reviewer’s suggestions and summarized our response in the following to address your concerns:
>
> 1. Weakness 1: The framework assumes that the Mamba-generated matrices adequately capture the dynamics of the underlying system. This assumption may not hold for highly nonlinear systems where matrix representations could become unstable or inaccurate.
>
> We thank the reviewer for the valuable comment. To further demonstrate the applicability of the proposed method on highly nonlinear systems, we evaluate the performance of MamKO on a large-scale nonlinear water treatment system [1] in F.4 of the Appendix, which contains 145 states, two control inputs, and 14 known disturbances. The dynamic behaviors of this process are simulated using a high-fidelity simulator that was built based on 145 ordinary differential equations (ODEs) [2].
> The modeling performance and the control performance are shown in Figure 16 and Figure 17 of the Appendix, respectively.
> From the corresponding experimental results, the proposed method outperforms the deep Koopman operator (DKO) and multilayer perception (MLP) methods in modeling tasks. Additionally, in a set-point tracking task, the MamKO-based model predictive controller (MPC) exhibited superior control performance compared to SAC, DKO-based MPC, and MLP-based MPC. From the experimental results, the proposed MamKO-based MPC reduced tracking error by 55.60%, 45.95%, and 10.07% as compared to MLP-based MPC, DKO-based MPC, and SAC, respectively.
> The results demonstrate that the system is applicable to highly nonlinear systems.
>
> Following the comment as well as Comment #2 from Reviewer 41k8, we have added the large-scale nonlinear water treatment system simulation in F.4 of the Appendix to evaluate the modeling and control performance of the proposed method for high nonlinear systems.
>
> 2. Weakness 2: Experimental results do support the claims of improved performance. However, the paper could enhance its rigor by including statistical analyses (e.g., confidence intervals or hypothesis tests) to demonstrate the significance of the results.
>
> We appreciate the reviewer's insightful comment. The standard deviations of test loss in modeling tasks and the cost in control tasks, which are calculated based on ten trials of different initial conditions, are presented in Figures 2 and 3 of the revised manuscript. These results are helpful for estimating the confidence intervals of the modeling and control performance. The results demonstrate consistent modeling and control performance of the proposed method and confirm that the proposed method can provide superior control performance in comparison to the baseline methods.
>
> Following the comment, we have slightly changed the captions of Figure 2 and Figure 3 in the revised manuscript to stress the confidence intervals of the proposed method.
>
> 3. Weakness 3:  Authors do acknowledge that their method may not generalize to all real-world systems, but a more explicit discussion on the boundaries of applicability (e.g., systems with extremely high-frequency dynamics) would add value.
>
> We thank the reviewer for the valuable comment.
> In this work, we consider systems with various time scales (i.e., systems with high-frequency dynamics and low-frequency dynamics).
> For the CartPole system, we consider a time-vary system parameter $\mu_t^c$, which is generated following a sinusoidal function of various frequencies, to test the control performance of the proposed methods. The variation in the frequency of the sinusoidal function leads to changes in the time scale of the system dynamics.
> The results indicate that as the frequency increases, the advantages of MamKO become more pronounced.
> Nonetheless, the applicability of our methods to systems with extremely high-frequency dynamics has not been considered in this work, which we plan to investigate in our future work.
>
> [1]  J. Alex, L. Benedetti, J. Copp, K. Gernaey, U. Jeppsson, I. Nopens, M. Pons, L. Rieger, C. Rosen, J. Steyer, et al. Benchmark simulation model No. 1 (BSM1). Report by the IWA Task group on benchmarking of control strategies for WWTPs, 1, 2008.
>
> [2] J. Zeng and J. Liu. Economic model predictive control of wastewater treatment processes. Industrial & Engineering Chemistry Research, 54(21):5710–5721, 2015.

---

> > ### Author Response · Authors · 2024-11-28
> > **Responses to Reviewer A4oR**
> >
> > 4. Weakness 4:  Comparison: The benchmark methods used have all been solved using classical/modern control methods, yet the author did not include any non-data-driven solutions for comparison. This may be a useful addition to the paper.
> >
> > We thank the reviewer for the insightful suggestions.
> > For traditional non-data-driven controllers like (proportional–integral–derivative) PID controllers, the control performance can be excellent when parameters are finely tuned, making them widely preferred in the industry. However, unlike MPC, which explicitly represents constraints, guaranteeing constraints in PID controllers can be challenging. Additionally, tuning PID controllers for multi-input and multi-output systems can be quite complex.
> > Another practical method is the first-principles model-based MPC. As highlighted in the introduction, this approach requires considerable time and expertise to develop an accurate first-principles model, which is a significant drawback. Furthermore, this method can result in suboptimal solutions and impose a significant computational burden due to the need to solve non-convex optimization problems.
> >
> > We include both the PID controller and MPC based on the first-principles model as baselines of the water treatment process in F.4 of the Appendix. The control performance is presented in Figure 17 of the Appendix.
> > According to Figure 17 of the Appendix, our method achieves a reduction in tracking error of 11.56% compared to the PID controller. Furthermore, integrating state constraints with the PID controller for this system can be challenging, and switching to a different set-point may require retuning the parameters.
> > The MPC based on the exact first-principles model (the same model as the simulator) exhibits the smallest tracking error. However, as emphasized in the introduction, obtaining an exact first-principles model can be challenging. Additionally, the nonlinear optimization required for first-principles model systems often results in suboptimal solutions, leading to large deviation points in the process.
> >
> > In response to the comment, we have included both the PID controller and MPC based on the first-principles model as baselines of the water treatment process in F.4 of the Appendix, where the control performance is presented in Figure 17 of the Appendix.
> >
> > 5. Question 1: Recommended fixes: which provides enhanced model predictability and adaptability, as compared to Koopman models with constant Koopman operators -- you mean prediction capability ? Since time scales are important, figures should include time , instead of time steps ( or epochs), Fig 8 just has ``t'' with no units, Please provide unit on all figure axes where possible.
> >
> > We appreciate the reviewer's suggestions.
> > For the model predictability, we thank the reviewer for pointing out the typo, and we have revised the
> > predictability to prediction capability in the Abstract, which is a more rigorous expression.
> > For the suggestion on the units, we have revised all the figures and added units to improve clarity. To better present the time scale of each system, we have included Table 3 in the revised manuscript to present sampling periods of the benchmark systems.
> >
> > Following the comment, we have revised the word predictability to prediction capability in the Abstract.
> > We have added the units for each figure and included the sampling periods of the benchmark systems in Table 3 of the revised manuscript.

---

> > > ### Author Response · Authors · 2024-11-28
> > > **Responses to Reviewer A4oR**
> > >
> > > 6. Question 2: Claims requiring more evidence:
> > > MamKO achieves good balance between advanced modeling capabilities and real-time control implementation efficiency $-->$ This claim in RT implementation needs more support hence actual time units are needed. The RSCP problem has a scale of hours and is therefore generally very slow. It will be good to see clearly the range of problems in time scale and frequency scale to understanding capability against system dynamics.
> > >
> > > We thank the reviewer for the suggestion. MamKO, by leveraging the concept of the Koopman operator [1], can achieve high computational efficiency with the convex optimization from the linear model.
> > > We evaluate the computation times of the control methods on different benchmark systems. The online control implementation of the control methods is conducted on a computer equipped with an Intel Core i9-13900K CPU and 128 GB DDR4 RAM. The computation times of the MamKO-based MPC and baselines during control on the benchmark systems are presented in Table 2 of the revised manuscript. Compared with the MLP-based MPC, the MamKO-based MPC reduces the computation time by 98.38%, 83.39%, 99.21%, 90.34%, and 99.17% for the five benchmark systems, respectively. To further demonstrate the computational efficiency of the proposed method, the sampling periods of the systems are presented in Table 3 of the revised manuscript. A comparison of the sampling periods and computation times demonstrates that the proposed MamKO-based MPC method can reliably ensure real-time implementation for each of the considered systems.  The efficient real-time implementation of the proposed method stems from the use of a linear state-space model within the proposed framework, which facilitates the formulation of an optimal control problem that requires solving convex optimization despite the nonlinearity in the dynamics of the considered systems.
> > >
> > > Following the reviewer's comment, as well as Comment #3 from Reviewer 9kPH and Comment #6 from
> > > Reviewer 41k8, we have added Table 2 in the revised manuscript, which presents the computation times for each method across the benchmark systems. Additionally, we have added Table 3 to provide the sampling periods of the benchmark systems.
> > >
> > > 7. Question 3: Theoretical: The paper could use more rigorous theoretical analysis or proofs to back the claims of stability and effectiveness in capturing time-varying dynamics. For example, formal proofs of the convergence properties of the Mamba-based Koopman operators or bounds on the error in approximating nonlinear systems would greatly increase the paper’s credibility.
> > >
> > > We thank the reviewer for the insightful comment.
> > > The primary objective is to leverage the integration of the Mamba and Koopman operators to develop a unified modeling and control approach for general nonlinear systems with time-varying parameters. We agree with the reviewer that a formal theoretical analysis of the convergence or the error bounds of the proposed approach will be instrumental in ensuring the proposed method can achieve the desired prediction performance to capture the time-varying dynamics. Meanwhile, the convergence analysis for the proposed control method, which is built based on time-varying Koopman operators, is challenging due to several factors. First, the underlying nonlinear system features time-varying parameters. Second, the use of time-varying Koopman operators inside the Mamba-based structure further complicates the theoretical analysis. Therefore, formal theoretical analysis of convergence is left to our future work.
> > >
> > > In response to this comment, as well as Comment #1 from Reviewer PG54 and Comment #3 from Reviewer 41k8, we have added a sentence at the end of Section 6 of the revised manuscript to discuss this future research direction.
> > >
> > > [1] B. O. Koopman. Hamiltonian systems and transformation in Hilbert space. Proceedings of the National Academy of Sciences, 17(5):315–318, 1931.

---

> > > > ### Author Response · Authors · 2024-11-28
> > > > **Responses to Reviewer A4oR**
> > > >
> > > > 8. Question 4: Equation (5): The transition from a time-invariant to a time-varying state-space model is theoretically justified but lacks a rigorous mathematical proof. The paper relies heavily on empirical evidence, perhaps some theoretical discussion may be help support the generalizability of the method.
> > > >
> > > > We appreciate the reviewer’s insightful suggestion.
> > > > At a given time instant $k$, the time-varying nonlinear system in Eq. (4) of the revised manuscript can be regarded as a time-invariant system in Eq. (1) of the revised manuscript. Thus, the dynamics at each time instant can be represented by the Koopman model in Eq. (3) of the revised manuscript. This implies that a distinct set of matrices {$A, B, C$} can capture the dynamics at a specific time instant. However, as time proceeds, the matrices {$A, B, C$} for the previous instant will not be sufficient for describing the dynamics at the current instant. An effective approach to model this time-varying system involves identifying a set of matrices {$A_k, B_k, C_k$} for each time instant $k$, which adapts to the changing dynamics. The solution can capture the dynamics of the system for each time instant, leading to the formulation of the time-varying Koopman model in Eq. (5) of the revised manuscript.
> > > >
> > > > Following the comment of the reviewer, we have included the above discussion in the second paragraph of Section 3.2 of the revised manuscript.
> > > >
> > > > 9. Question 5: CELU Activation: The use of the negative CELU function is an interesting choice to handle instability. The mathematical justification provided for this choice seems reasonable, but more in-depth analysis of how it impacts model stability across different systems would strengthen the claim.
> > > >
> > > > We thank the reviewer for the suggestion. To find the activation to bound the eigenvalues, we considered several options, including the Gaussian Error Linear Unit (GELU), Exponential Linear Unit (ELU), Sigmoid Linear Unit (SiLU), and Continuously Differentiable Exponential Linear Unit (CELU). Through trial and error, CELU demonstrated the best modeling performance across all three tested systems. Also, we conduct an evaluation in Section 5.3, which can also prove the superiority of the CELU.
> > > >
> > > > For the theoretical analysis, the connection between the stability of the original system and the established model is an insightful topic where the choice of activation is important. For linear state-space models, the stability of the system can be analyzed by the eigenvalue of matrix $A$.
> > > > However, the linear state-space models used in our study are derived from nonlinear transformations by neural networks. In the MamKO model, linear state-space model is generated with historical data,
> > > > which also complicates the theoretical proof of the stability analysis of the learned system.
> > > > Overall, due to the complicated nature of the structures involved, conducting a theoretical analysis of how the activation function influences eigenvalues presents significant challenges. The rigorous analysis is beyond the scope of this paper, but we intend to explore this topic in future research.
> > > >
> > > > We thank the reviewer again for all the comments.

---

### Official Review · Reviewer_9kPH · 2024-11-03

**Soundness:** 3
**Presentation:** 4
**Contribution:** 3
**Rating:** 8
**Confidence:** 3

**Summary:**

In this paper, the authors propose MamKo, combining Koopman operator theory (projecting nonlinear systems into a high-dimensional linear space) with Mamba (a LLM based on state-space models) for accurate and efficient modeling of time-varying systems. Furthermore, based on the derived model, a model-based control approach is proposed. Simulations with different, time-varying systems demonstrate that MamKo can be superior to other SOTA methods.

The main contribution is combining these two methods (with some minor modifications), creating a powerful framework for time-varying systems.

**Strengths:**

- The paper is well written, the ideas are clearly formulated, and the results are well presented.
- To the best of my knowledge, it is the first time that these two methods have been combined and applied to model time-varying systems. Thus, it's a novel and original approach.
- The simulation results are quite convincing that the proposed method can significantly improve the modeling and control of time-varying systems.

**Weaknesses:**

- The simulation is limited to learning method but ignores "classic" system identification. There is a rich literature on identifying time-varying parameters in dynamical systems. In particular, for the cart pendulum example, a comparison to classic system identification would strengthen the paper.
- Maybe I missed the part but how is the optimal dimension of the linear space, i.e. the dimension of A_k in (5) determined?
- In the introduction, the computational time of other koopman based approaches is discussed. However, there is no proper discussion/evaluation/theory on the training/inference time with the proposed framework.

**Questions:**

See weaknesses

---

> ### Author Response · Authors · 2024-11-28
> **Responses to Reviewer 9kPH**
>
> Thank you very much for the detailed comments, which are very helpful to us. We have revised the manuscript according to the reviewer’s comments and provided our responses as follows:
>
> 1. Weakness 1: The simulation is limited to learning method but ignores “classic'' system identification. There is a rich literature on identifying time-varying parameters in dynamical systems. In particular, for the cart pendulum example, a comparison to classic system identification would strengthen the paper.
>
> We thank the reviewer for suggesting ways to strengthen the paper. Following this comment, we apply a classical system identification method -- subspace identification [1] to the CartPole system. The open-loop prediction results are shown in Figure 7 of the Appendix.
> For the 30-step-ahead prediction task, the model built based on subspace identification can only provide satisfactory predictions of the position and velocity of the cart, while the predictions of the state variables related to the pole diverge. In contrast, the proposed method can accurately forecast the future behaviors of all the states of the CartPole system.
>
> Following the comment, we have added the experimental results in Figure 7 of the Appendix to show the modeling results from the MamKO method and the subspace identification method.
>
> 2. Weakness 2: Maybe I missed the part but how is the optimal dimension of the linear space, i.e. the dimension of $A_k$ in (5) determined?
>
> We thank the reviewer for the valuable comment. In our evaluation, the dimension of the linear state-space model is determined through trial and error, with the aim of achieving a balance between computational efficiency and modeling performance.
>
> We would like to use the case study on the CartPole to illustrate the selection of the lifting dimension of the state-space model. The results, which include the test loss across various lifting dimensions, are presented in Table 7 of the Appendix.
> As the dimension of the lifted space increases from 5 to 8, the test loss decreases significantly to $6.93\times 10^{-4}$. Further increasing the dimension does not lead to significant improvement in the modeling performance.
>
> Following the comment, we have included results in Table 7 that explore the relationship between test loss and the lifting dimension of the CartPole system.
>
> 3. Weakness 3: In the introduction, the computational time of other Koopman-based approaches is discussed. However, there is no proper discussion/evaluation/theory on the training/inference time with the proposed framework.
>
> We thank the reviewer for the valuable comment. MamKO, by leveraging the concept of the Koopman operator [2], can achieve high computational efficiency in both the prediction of future dynamic behavior and online optimal control for nonlinear systems. The training process of the MamKO method is implemented offline with the collected data, which does not directly influence the online implementation of the method. For the inference time, we evaluate the computation times of the control methods on different benchmark systems. The online control implementation of the control methods is conducted on a computer equipped with an Intel Core i9-13900K CPU and 128 GB DDR4 RAM. The computation times of the MamKO-based MPC and baselines during control on the benchmark systems are presented in Table 2 of the revised manuscript. Compared with the MLP-based MPC, the MamKO-based MPC reduces the computation time by 98.38%, 83.39%, 99.21%, 90.34%, and 99.17% for the five benchmark systems, respectively. To further demonstrate the computational efficiency of the proposed method, the sampling periods of the systems are presented in Table 3 of the revised manuscript. A comparison of the sampling periods and computation times demonstrates that the proposed MamKO-based MPC method can reliably ensure online implementation for each of the considered systems.  The efficient online implementation of the proposed method stems from the use of a linear state-space model within the proposed framework, which facilitates the formulation of an optimal control problem that requires solving convex optimization despite the nonlinearity in the dynamics of the considered systems.
>
> Following the comment  as well as Comment #6 from Reviewer A4oR and Comment #6 from
> Reviewer 41k8, we have included Table 2 of the revised manuscript to show the computation time of each method in the benchmark systems and Table 3 of the revised manuscript to present sampling periods of the benchmark systems. We have discussed the practicability of the MamKO-based MPC for real-time implementation on benchmark systems in Section 5.5 of the revised manuscript.
>
> [1] V. Overschee and B. De Moor. Subspace Identification of Linear Systems: Theory, Implementation,
> Applications. Springer Publishing, 1996
>
> [2] B. O. Koopman. Hamiltonian systems and transformation in Hilbert space. Proceedings of the National
> Academy of Sciences, 17(5):315–318, 1931.

---

### Meta-Review · Area_Chair_3ZAT · 2024-12-21

**Metareview:**

This paper introduces MamKO, a framework combining the Mamba architecture with Koopman operators to model and control nonlinear, time-varying systems effectively. The authors demonstrate MamKO's ability to generate adaptive Koopman operators from real-time data, enhancing modeling accuracy and real-time control efficiency, particularly in model predictive control (MPC) tasks. Strengths include its innovative integration of Mamba with Koopman operators, comprehensive experimental validation across various benchmarks, computational efficiency, and robustness. Weaknesses include limited theoretical guarantees for stability and convergence, lack of hardware experiments, and missing comparisons to alternative sequence models. Despite these, the paper presents a significant step forward in leveraging large language models for control applications, and the thorough rebuttal effectively addresses reviewers’ concerns, supporting its acceptance.

**Additional Comments On Reviewer Discussion:**

During the rebuttal, reviewers raised concerns about theoretical guarantees, hardware validation, error accumulation, and broader experimental comparisons. The authors responded by adding detailed discussions on error propagation, robustness, and computational efficiency. They included new experiments on time-delay systems and trajectory tracking tasks while demonstrating MamKO's performance on a large-scale nonlinear system. While theoretical guarantees and hardware validation remain future work, the authors clarified their rationale for choosing Mamba and expanded on evaluation metrics. The rebuttal effectively addressed key concerns, demonstrating the framework's robustness and practical potential, justifying its acceptance.

---

### Decision · Program_Chairs · 2025-01-22

Accept (Spotlight)